# Induction of a transcriptional adaptation response by RNA destabilization events

Lihan Xie[1,8], Gabrielius Jakutis[1,8], Christopher M Dooley [ID][1,8], Stefan Guenther[2], Zacharias Kontarakis [ID][1,5,6], Sarah P Howard [ID][1], Thomas Juan [ID][1,3,4,7✉] & Didier Y R Stainier [ID][1,3,4✉]

## Abstract

**Transcriptional adaptation (TA) is a cellular process whereby mRNA-destabilizing mutations are associated with the transcriptional upregulation of so-called adapting genes. The nature of the TA-triggering factor(s) remains unclear, namely whether an mRNA-borne premature termination codon or the subsequent mRNA decay process, and/or its products, elicits TA. Here, working with mouse *Actg1*, we first establish two types of perturbations that lead to mRNA destabilization: Cas9-induced mutations predicted to lead to mutant mRNA decay, and Cas13d-mediated mRNA cleavage. We find that both types of perturbations are effective in degrading *Actg1* mRNA, and that they both upregulate *Actg2*. Notably, increased chromatin accessibility at the *Actg2* locus was observed only in the Cas9-induced mutant cells but not in the Cas13d-targeted cells, suggesting that chromatin remodeling is not required for *Actg2* upregulation. We further show that ribozyme-mediated *Actg1* pre-mRNA cleavage also leads to a robust upregulation of *Actg2*, and that this upregulation is again independent of chromatin remodeling. Together, these data highlight the critical role of RNA destabilization events as a trigger for TA, or at least a TA-like response.**

**Keywords** CRISPR/Cas9; CRISPR/Cas13d; mRNA Decay; Self-cleaving Ribozyme; Transcriptional Adaptation
**Subject Categories** Chromatin, Transcription & Genomics; RNA Biology

## Introduction

Transcriptional adaptation (TA), a cellular process whereby mRNA-destabilizing mutations lead to the transcriptional upregulation of specific genes, termed adapting genes (Rossi et al, 2015; El-Brolosy et al, 2019; Ma et al, 2019), lies at the intersection of two conserved gene regulation pathways: mRNA surveillance (Houseley and Tollervey, 2009) and small RNA-mediated gene regulation (Morris and Mattick, 2014). While initially discovered in zebrafish (*Danio rerio*) (Rossi et al, 2015; Sztal et al, 2018; Ma et al, 2019; Diofano et al, 2020; Kürekçi et al, 2021; Pomreinke and Muller, 2024), evidence for TA now exists across multiple species from *C. elegans* (Serobyan et al, 2020; Fernandez-Abascal et al, 2022; Welker et al, 2023) and Silkworm (*Bombyx mori*) (Wang et al, 2024) to mouse (El-Brolosy et al, 2019; Jiang et al, 2022) and human (Falcucci et al, 2025). However, the nature of the TA-triggering factor(s) remains under debate.

Our data in zebrafish (El-Brolosy et al, 2019), *C. elegans* (Serobyan et al, 2020), mouse cells in culture (El-Brolosy et al, 2019), and human cells in culture (Falcucci et al, 2025) have shown that loss of UPF1, or its orthologous protein, an essential component of the nonsense-mediated mRNA decay (NMD) pathway (Kim and Maquat, 2019), at least partially blocks TA. This and other observations, including the finding that RNA-less alleles do not exhibit TA, led us to hypothesize that mRNA fragments, presumed products of NMD, translocate into the nucleus and modulate gene expression. Consistent with this hypothesis, we identified in *C. elegans* a 25 bp sequence sufficient to elicit the TA response in the *act-5/act-3* model (Welker et al, 2023).

On the other hand, Ma et al (2019) proposed a competition-based model where Upf3a binds to full-length mRNAs that contain a premature termination codon (PTC) and recruits H3K4me3 'writers' within the nucleus to the adapting gene locus/loci, whereas the Upf3b-bound mutant mRNAs are recognized by Upf1 and Upf2 for NMD. In a subsequent study, the same group reported that Upf3a but not Upf1 is required for the genetic compensation response in a *leg1a–leg1b* zebrafish model (Xie et al, 2023), further distinguishing the two proposed mechanisms (i.e., mRNA decay versus Upf3a binding).

Recent advances in DNA and RNA engineering have expanded the toolbox for controlling the fate of RNAs and prompted us into new attempts at identifying the TA-triggering factors via bypassing PTCs and their associated proteins. In this study, we use the RNA-targeting CRISPR/Cas13d from *Ruminococcus flavefaciens* (Yan et al, 2018) as well as a self-cleaving T3H48 ribozyme variant from

[1]Department of Developmental Genetics, Max Planck Institute for Heart and Lung Research, Bad Nauheim, Hessen 61231, Germany. [2]ECCPS Bioinformatics and Deep Sequencing Platform, Max Planck Institute for Heart and Lung Research, Bad Nauheim, Hessen 61231, Germany. [3]German Centre for Cardiovascular Research (DZHK), Partner Site Rhine-Main, Bad Nauheim, Hessen 61231, Germany. [4]Excellence Cluster Cardio-Pulmonary Institute (CPI), Bad Nauheim, Giessen, Frankfurt, Germany. [5]Present address: Genome Engineering and Measurement Laboratory (GEML), Eidgenössische Technische Hochschule (ETH) Zürich, Zürich, Switzerland. [6]Present address: Functional Genomics Center Zürich, ETH Zürich/University of Zürich, Zürich 8057, Switzerland. [7]Present address: Department of Immunology, Genetics and Pathology, Uppsala University, Uppsala 75 185, Sweden. [8]These authors contributed equally: Lihan Xie, Gabrielius Jakutis, Christopher M Dooley. ✉E-mail: Thomas.Juan@igp.uu.se; Didier.Stainier@mpi-bn.mpg.de

*Schistosoma mansoni* (Zhong et al, 2020) to achieve robust RNA cleavage of mouse *Actg1*. We have previously reported TA in mouse *Actg1* mutant cells, with *Actg2* as an adapting gene (El-Brolosy et al, 2019). Here, we compare mRNA-decay displaying *Actg1* mutant alleles, *Actg1* mRNA targeting by Cas13d, and *Actg1* pre-mRNA cleavage by the T3H48 ribozyme, and show that all three approaches are sufficient to cause *Actg2* upregulation. These findings further highlight the role of RNA destabilization as a trigger for TA and further suggest that RNA decay rather than PTCs is crucial for this process.

## Results and discussion

### mRNA-destabilizing mutations in *Actg1* lead to transcriptional upregulation of *Actg2*

We have previously reported that in mouse NIH3T3 cells, an *Actg1* mutant allele lacking the endogenous stop codon, a so-called non-stop decay (NSD) allele, and exhibiting mutant mRNA decay, displays a more pronounced upregulation of *Actg2* than does an *Actg1* RNA-less allele (El-Brolosy et al, 2019). To further investigate this TA model, we generated, using CRISPR/Cas9 technology in NIH3T3 cells, two different *Actg1* nonsense alleles: one that contains a PTC in its third exon (denoted as *Actg1*^PTC1), and another that has in its fifth exon a 4 kb insertion that forms an NMD-prone long 3'UTR (denoted as *Actg1*^PTC2), as well as a new large deletion (LD) RNA-less allele (denoted as *Actg1*^LD) (Figs. 1A and EV1A). RT-qPCR analyses revealed significantly lower *Actg1* mRNA levels in *Actg1*^PTC1 and *Actg1*^PTC2 cells concomitant with *Actg2* upregulation at both the mRNA and pre-mRNA levels (Figs. 1B and EV1B). We also observed an upregulation of *Actg1* pre-mRNA levels in *Actg1*^PTC1 and *Actg1*^PTC2 cells (Figs. 1B and EV1B), suggesting that the mutant gene itself can be targeted by the TA machinery, consistent with our previous observations (El-Brolosy et al, 2019; Jiang et al, 2022). The magnitude of *Actg2* upregulation in *Actg1*^PTC1 and *Actg1*^PTC2 cells greatly exceeds that observed in *Actg1*^LD cells, consistent with our previous findings (El-Brolosy et al, 2019), further suggesting that *Actg2* upregulation results primarily from mRNA decay and not the loss of ACTG1 protein. These results further indicate that *Actg2* is an adapting gene in *Actg1* mutant cells that display mutant mRNA decay.

### *Actg2* upregulation in *Actg1* PTC mutants is associated with increased chromatin accessibility

Given that increased transcription is often associated with increased chromatin accessibility (Cao et al, 2018), we next performed an ATAC-seq analysis to evaluate the changes in chromatin accessibility across the genome. Focusing on the *Actg2* locus, we observed loosening of the chromatin in three regions: a 5' intergenic region (−6.7 to −5.5 kb from the transcription start site (TSS)), around the TSS (−0.3 to +0.6 kb), and in the first intron (2.2 to 2.5 kb), which were also open in wild-type cells, albeit to a much lower extent (Figs. 1C and EV1C). Importantly, these changes were only observed in *Actg1*^PTC1 and *Actg1*^PTC2 cells and not in *Actg1*^LD cells (Figs. 1C and EV1C), indicating that TA, or its consequences, can lead to chromatin remodeling in *Actg1* mutants that display mutant mRNA decay, independently of a protein feedback effect.

### Cas13d-mediated *Actg1* mRNA cleavage leads to transcriptional upregulation of *Actg2*

Kushawah et al (2020) have previously shown that Cas13d can degrade specific mRNAs in zebrafish embryos and that *vclb* mRNA levels increase after the cleavage of *vcla* mRNA, similar to our observations in *vcla* mutant zebrafish that display mutant mRNA decay (El-Brolosy et al, 2019). Therefore, we leveraged CRISPR/Cas13d as another means of triggering mRNA decay and tested whether it also leads to transcriptional modulation in mouse cells in culture. Using plasmids constructed by the Hsu laboratory (Konermann et al, 2018), we first generated an NIH3T3 cell line with a stable integration of an EF-1alpha promoter-driven *Cas13d* in the *Hipp11* safe harbor locus (Hippenmeyer et al, 2010). We then transfected these cells with a Cas13d guide RNA (gRNA) designed to specifically target *Actg1* mRNA (Fig. 2A, top) or *GFP* mRNA as a negative control. Cas13d-mediated *Actg1* mRNA cleavage resulted in a significant decrease in *Actg1* mRNA levels at 14 h after gRNA transfection (Fig. 2A, bottom), in line with previous Cas13d-mediated knockdown efficiency data in zebrafish embryos (Kushawah et al, 2020), and in mouse and human cells in culture (Morelli et al, 2023). We also observed a significant upregulation of *Actg1* pre-mRNA levels as well as *Actg2* mRNA and pre-mRNA levels upon *Actg1* mRNA cleavage by Cas13d (Fig. 2A, bottom). Furthermore, we assessed ACTG1 protein levels in the same conditions by western blot analysis and observed no significant difference between cells transfected with the *GFP* gRNA or the *Actg1* gRNA (Figs. 2B and EV2), suggesting that in these experiments *Actg2* upregulation is not due to reduced ACTG1 protein levels. These results indicate that cytoplasmic mRNA cleavage/decay without a genomic lesion is sufficient to induce TA/TA-like responses.

Next, we tested two other models selected based on literature review, namely *Ctnna1*, since *Catenin* mutants in mouse display the upregulation of their paralogs in several tissues (Stocker and Chenn, 2006; Li et al, 2012; Vite et al, 2018) and *Nckap1*, an autism-associated gene that when knocked out in B16-F1 melanoma cells induces *Nckap1l* upregulation and partial functional compensation (Dolati et al, 2018). Consistent with the previous results for *Actg1*, we observed efficient decay of *Ctnna1* and *Nckap1* mRNA after Cas13d-mediated cleavage as well as significant upregulation of several paralogs (Fig. 2C,D): cleavage of *Ctnna1* mRNA resulted in higher levels of *Ctnna2* and *Ctnna3* mRNA (Fig. 2C), and cleavage of *Nckap1* mRNA resulted in higher levels of *Nckap1l*, *Nckap5*, and *Nckap5l* mRNA (Fig. 2D), suggesting that the Cas13d-mediated mRNA cleavage-induced TA-like response is not limited to the *Actin* gene family.

### Cas13d-mediated *Actg1* mRNA cleavage leads to *Actg2* upregulation independently of chromatin accessibility changes

We next investigated whether Cas13d-mediated *Actg1* mRNA cleavage leads to chromatin remodeling at the *Actg2* locus, as we had observed in the *Actg1*^PTC cells (Figs. 1C and EV1C). To this end, we treated Cas13d-expressing NIH3T3 cells with two rounds of transfection of the *Actg1*-specific gRNA or *GFP* gRNA and maintained the Cas13d-mediated cleavage of *Actg1* mRNA for 5 days, allowing approximately 5 cell cycles to occur under constant

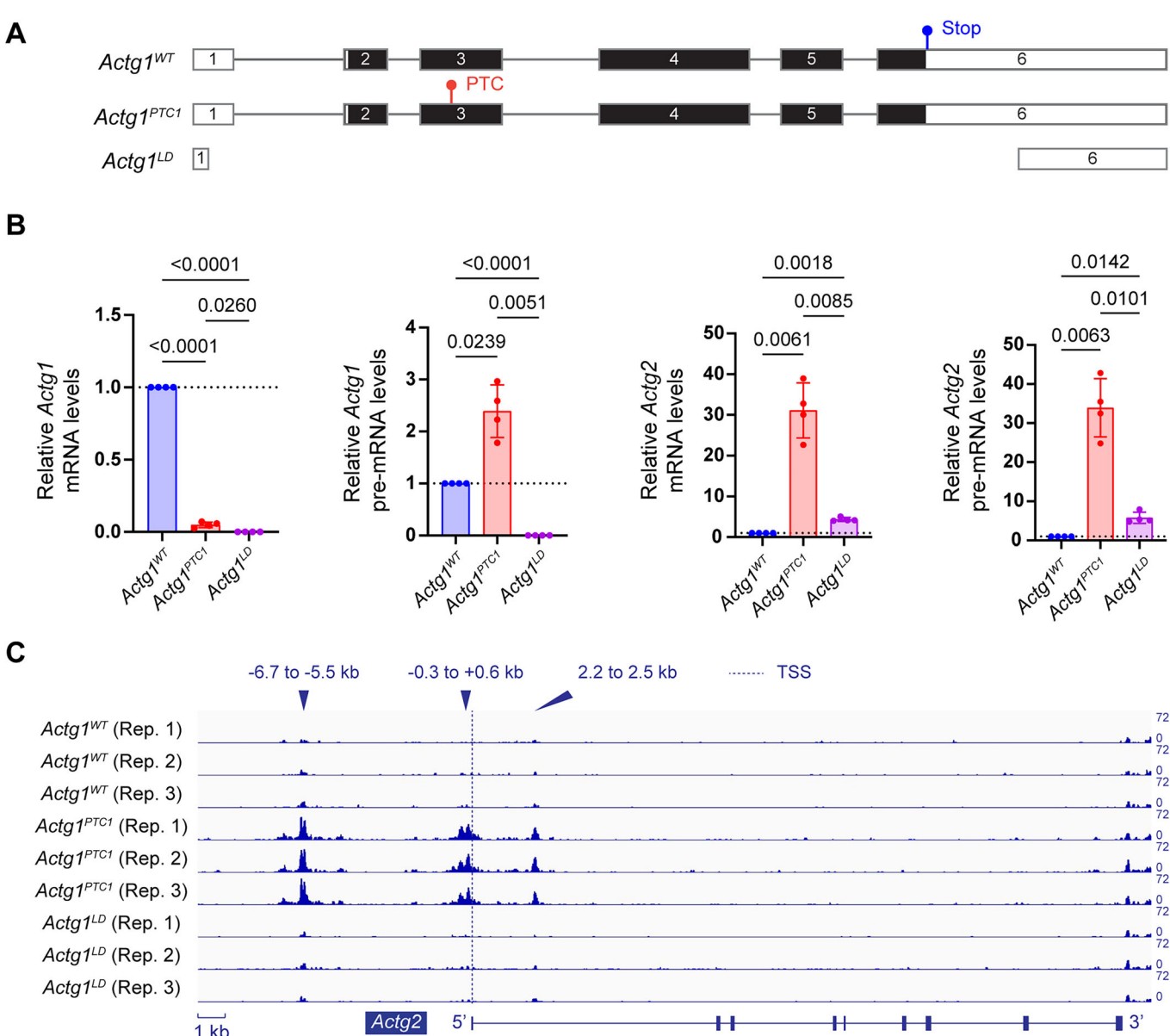

**Figure 1. Cas9-induced *Actg1* mutation leads to mutant mRNA decay and *Actg2* upregulation.**

(A) Schematic view of wild-type and mutant *Actg1* alleles. *Actg1^PTC1^* cells contain a premature termination codon (PTC) in exon 3; this PTC is located 45 and 73 bases from the 5′ and 3′ ends of the exon, respectively. *Actg1^LD^* cells contain three different lesions: (1) complete absence of the gene, (2) absence of the gene except for the first 44 bases of the 5′UTR, and (3) absence of the gene except for the last 441 bases of the 3′UTR. Detailed information about the genotype of *Actg1^PTC1^* and *Actg1^LD^* cells can be found in the Materials and Methods. (B) Relative mRNA and pre-mRNA levels of *Actg1* and *Actg2*. $n = 4$ biologically independent samples, one-way ANNOVA, pairwise comparison, and exact $p$ values are represented in the figure. Data are presented as mean ± standard deviation. (C) Chromatin accessibility at the *Actg2* locus. ATAC-seq analysis reveals three open chromatin regions in the *Actg1^PTC1^* allele located (1) in the 5′ intergenic region (i.e., −6.7 to −5.5 kb upstream of the transcription start site (TSS)), (2) around the TSS (i.e., −0.3 to +0.6 kb), and (3) in the first intron (i.e., 2.2 to 2.5 kb downstream of the TSS) of *Actg2*; $n = 3$ biologically independent samples. Source data are available online for this figure.

*Actg1* mRNA decay. We then performed ATAC-seq to evaluate the chromatin state at the *Actg2* locus and compared it with the previously identified peaks at and near the *Actg2* TSS in the *Actg1^PTC^* cells. Notably, ATAC-seq analysis did not identify any peaks that would correspond to regions of open chromatin at the *Actg2* locus (Fig. 2E), indicating that the Cas13d-mediated TA-like response does not require a significant extent of chromatin remodeling.

The difference in chromatin architecture manifested during Cas9- versus Cas13d-induced TA/TA-like responses might be due to the fact that *Actg1^PTC^* cells have experienced permanent mutant mRNA decay for a period of at least 28 days, thereby adopting a steady-state increase in *Actg2* transcription, whereas the Cas13d approach involves temporary *Actg1* mRNA perturbation events for 5 days. A previous report has shown that 5 days was sufficient for chromatin remodeling to occur under a continuous trigger (Rao

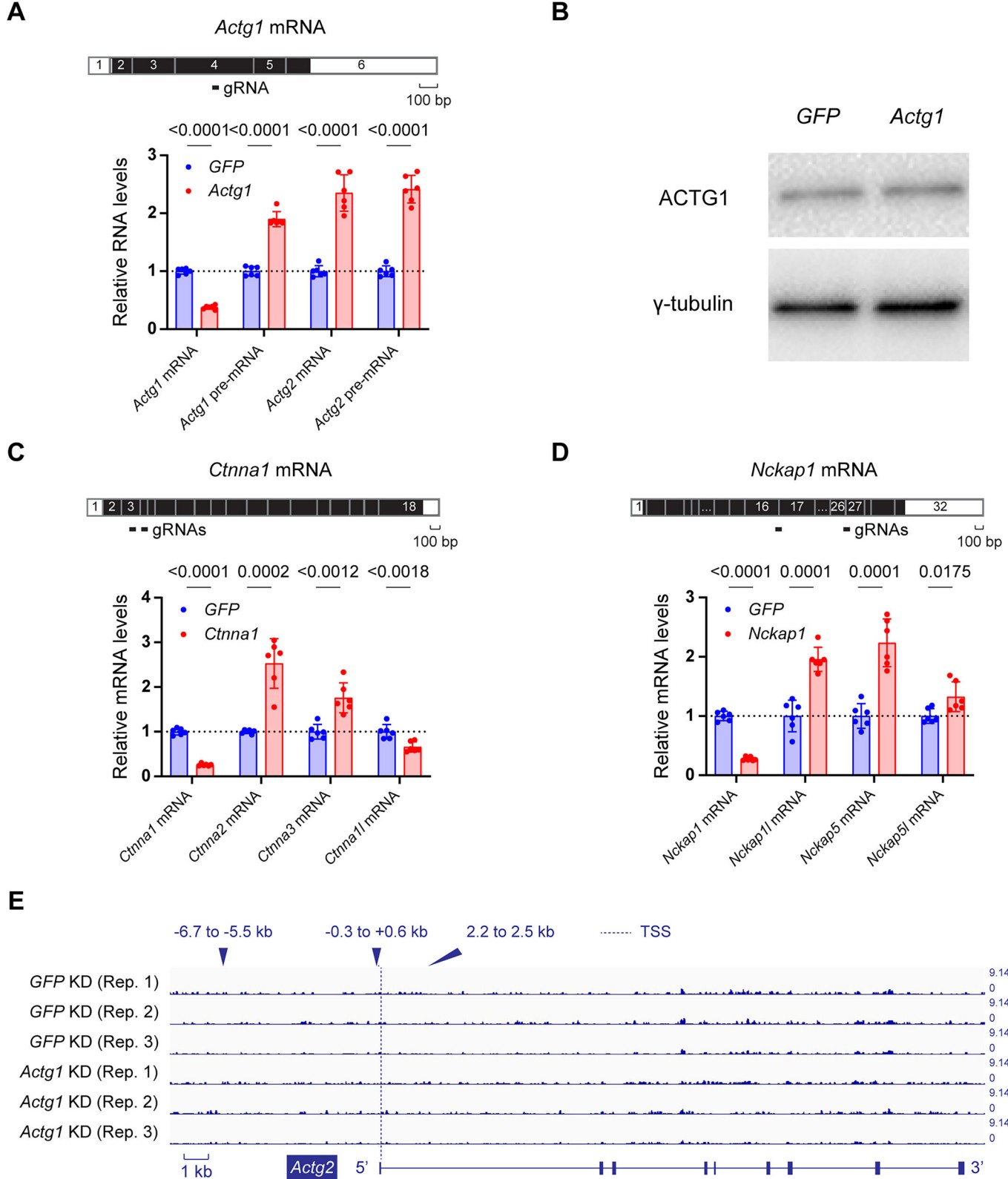

◀ **Figure 2.  Cas13d-mediated *Actg1* mRNA cleavage leads to *Actg2* upregulation.**

(A) Top: position of Cas13d gRNA targeting *Actg1* coding sequence. Bottom: Cas13d-mediated *Actg1* mRNA cleavage leads to decreased levels of *Actg1* mRNA, as well as increased levels of *Actg1* pre-mRNA, *Actg2* mRNA, and *Actg2* pre-mRNA. $n = 6$ biologically independent samples, unpaired t-test, and exact p values are represented in the figure. Data are presented as mean ± standard deviation. (B) Western blot analysis of the samples shown in Fig. 2A (150 ng of gRNA used; 14 h of transfection) reveals no obvious difference in ACTG1 protein levels between cells transfected with *GFP*- or *Actg1*-targeting gRNAs; γ-tubulin as loading control; full blot shown in Fig. EV2. (C, D) Top: position of Cas13d gRNAs targeting the coding sequence of *Ctnna1* (C) and *Nckap1* (D), respectively. Bottom: Cas13d-mediated mRNA cleavage leads to decreased mRNA levels of the targeted gene as well as increased mRNA levels of potential adapting genes: *Ctnna2* and *Ctnna3* for *Ctnna1* (C) and *Nckap1l*, *Nckap5*, and *Nckap5l* for *Nckap1* (D). $n = 6$ biologically independent samples, unpaired t-test, and exact p values are represented in the figure. Data are presented as mean ± standard deviation. (E) Chromatin accessibility at the *Actg2* locus remains unchanged after two rounds of gRNA transfection 5 days and 3 days before sample collection (see methods); position of the open chromatin peaks observed in *Actg1^PTC1* mutant cells marked as in Fig. 1C. $n = 3$ biologically independent samples. Source data are available online for this figure.

et al, 2001); however, examples have been reported during fly and worm development where rapid gene activation and deactivation occur in the absence of canonical histone modifications (Perez-Lluch et al, 2015). It is thus possible that chromatin accessibility changes occur over multiple cell divisions and are associated with the formation of epigenetic memory (Jiang et al, 2022); yet, they may be dispensable for immediate transcriptional modulation during TA/TA-like responses.

## Nuclear-localized Cas13d does not appear to induce significant nuclear RNA decay

Given that mRNA destabilization might play a crucial role during TA, based on our previous and current results, it is plausible that mRNA fragments, and/or their derivatives, enter the nucleus where they modulate adapting gene transcription through yet unknown mechanisms. This hypothesis led us to examine whether pre-mRNA decay in the nucleus could also lead to TA/TA-like responses.

We thus generated NIH3T3 cell lines expressing Cas13d fused with a nuclear localization signal (NLS) (Kushawah et al, 2020) from the same *Hipp11* safe harbor locus. We then transfected these cells with a gRNA targeting the second intron of *Actg1* (Fig. EV3, top). Contrary to what we observed with cytoplasmic Cas13d and an exon-targeting gRNA (Fig. 2A), nuclear Cas13d-NLS with an intron-targeting gRNA did not affect *Actg1* pre-mRNA or mRNA levels significantly, an observation consistent across four independent Cas13d-NLS knock-in cell lines (Fig. EV3, bottom). There are several possible reasons for this result including the mislocalization of Cas13d-NLS, the low efficiency of the intron-targeting gRNA, and the kinetics of RNA splicing versus those of Cas13d-NLS-mediated pre-mRNA cleavage. These results prompted us to explore alternative strategies to manipulate nuclear RNA stability.

## The self-cleaving T3H48 ribozyme destabilizes *Actg1* pre-mRNA and triggers *Actg2* upregulation

Hammerhead ribozymes (HHRs) are small, efficient, scissors-like RNAs that fold into designated structures and rapidly cleave themselves in *cis* (Doherty and Doudna, 2000). T3H48-HHR, an engineered variant of a natural ribozyme from *Schistosoma mansoni*, exhibits up to 99.9% knockdown efficiency in human cells for a luciferase reporter gene carrying T3H48-HHR in its 3' untranslated region (Zhong et al, 2020). Given the small size, fast kinetics, and high fidelity of T3H48-HHR, this tool serves as a good candidate to cleave and degrade newly transcribed pre-mRNAs in

the nucleus and determine the role of nuclear RNA decay during TA/TA-like responses.

Since the *Actg1* locus gives rise to multiple transcripts with non-canonical exon and intron compositions, we knocked in the 65 bp T3H48-HHR, insulated on both its 5' and 3' ends (Wurmthaler et al, 2019), into intron 3 as it is present in all *Actg1* transcripts (Fig. 3A). Notably, a single nucleotide substitution changes T3H48-HHR from a catalytically active form (denoted as T3H48-aHHR) to a catalytically inactive form (denoted as T3H48-iHHR) (Zhong et al, 2020), and we used the latter as a negative control. Genotyping was performed with two pairs of primers either within or flanking the knock-in cassette to select the homozygous T3H48-HHR knock-in cell lines (Fig. EV4A). Furthermore, the full-length *Actg1* cDNA was amplified and sequenced to test for any abnormal RNA splicing caused by the T3H48-HHR insertion (Fig. EV4B).

Then, we used RT-qPCR to assess the outcome of the T3H48-HHR knock-in and the consequent *Actg1* pre-mRNA cleavage in different clones, comparing them with wild-type, *Actg1^PTC1*, and *Actg1^LD* cells. As expected, T3H48-aHHR expression led to a significant decrease in *Actg1* mRNA levels and exhibited clone-specific knockdown efficiencies of 53.3% and 92.8%, respectively (Fig. 3B, left). The efficiency of *Actg1* pre-mRNA cleavage was inferred from the levels of *Actg1* mRNA detected in the *Actg1^aHHR1* and *Actg1^aHHR2* lines. By sequencing the *Actg1* cDNA in the different lines, we determined that the T3H48 HHR knock-in did not affect *Actg1* pre-mRNA splicing (Fig. EV4B), suggesting that the difference in *Actg1* mRNA levels between the *Actg1^aHHR1* and *Actg1^aHHR2* lines likely results from varying pre-mRNA cleavage efficiency. On the contrary, *Actg1* mRNA levels were similar in the *Actg1^iHHR* line as in wild type (Fig. 3B, left). We observed *Actg2* upregulation on both the mRNA and pre-mRNA levels, which correlated with the extent of *Actg1* knockdown by the T3H48^aHHR (Fig. 3B, middle and right), altogether suggesting that TA, or a TA-like response, is triggered by RNA decay taking place in the nucleus and/or cytoplasm. We then sought to compare the more efficient T3H48^aHHR knock-in allele (*Actg1^aHHR2*) with the *Actg1^PTC* and Cas13d knock-in alleles and investigate whether *Actg2* upregulation in the T3H48^aHHR-expressing cells was accompanied by local chromatin accessibility changes by performing an ATAC-seq analysis. We observed that chromatin accessibility at the *Actg2* locus remained unchanged in *Actg1^aHHR2* cells when compared with *Actg1^iHHR* cells (Fig. 3C) recapitulating the observations on the cells undergoing Cas13d-mediated *Actg1* mRNA cleavage.

In summary, we first confirmed that *Actg2* is an adapting gene in *Actg1* mutants that display mutant mRNA decay. Furthermore, we report that Cas13d-mediated mRNA cleavage of *Actg1* triggers the

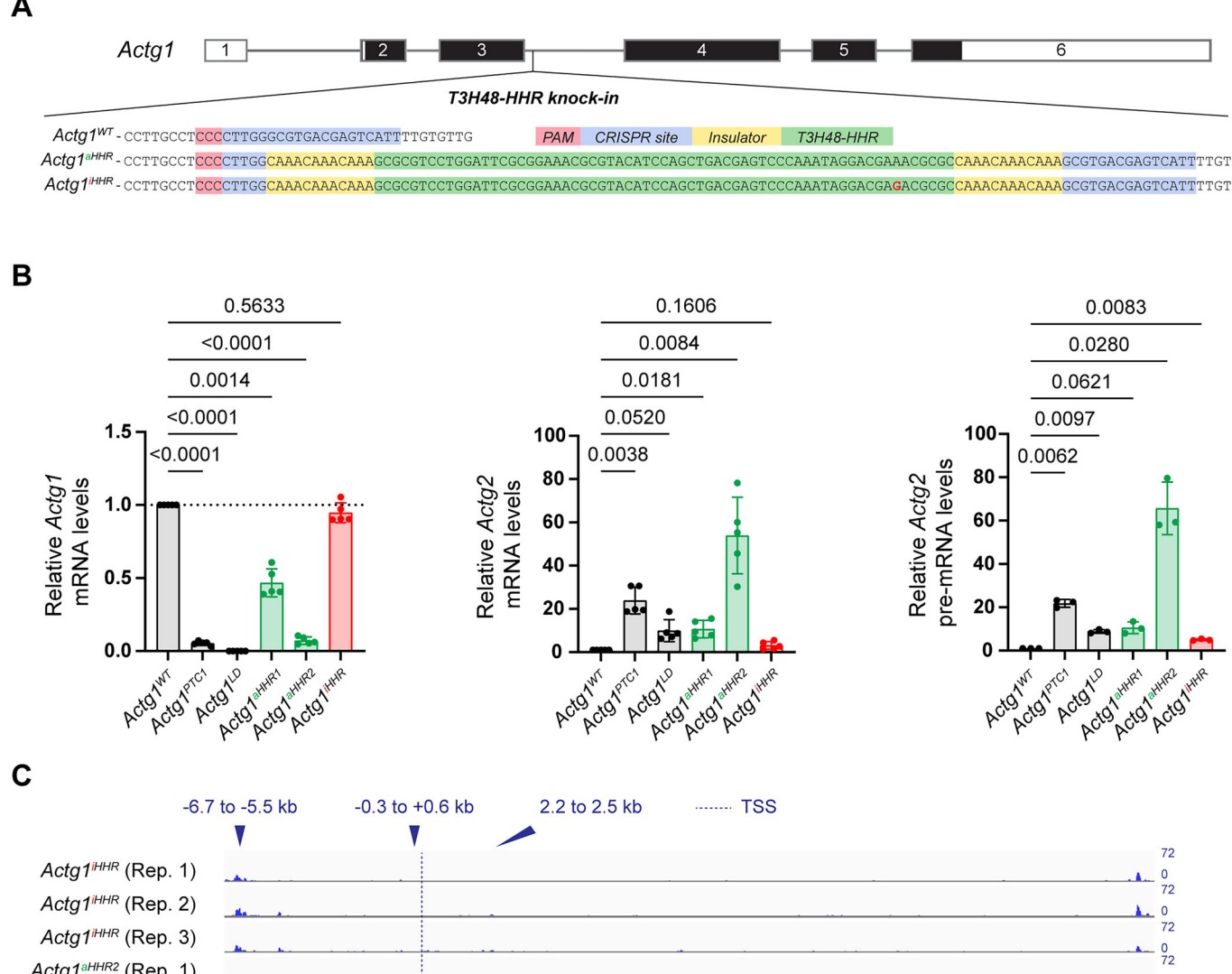

**Figure 3. Integration of a self-cleaving ribozyme in intron 3 of Actg1 leads to Actg1 pre-mRNA decay and Actg2 upregulation.**

(A) Schematic view of Actg1 wild-type and T3H48 hammerhead ribozyme (HHR) knock-in alleles. Actg1^aHHR and Actg1^iHHR cells harbor an insulated T3H48-HHR, located 33 and 253 bases from the 5' and 3' ends of intron 3, respectively. A point mutation (A > G, highlighted in red) in Actg1^iHHR cells renders the T3H48-HHR catalytically inactive. (B) Relative levels of Actg1 mRNA (left), Actg2 mRNA (middle), and Actg2 pre-mRNA (right) indicate Actg1 decay and Actg2 upregulation upon cleavage of Actg1 pre-mRNA by the T3H48 ribozyme. n = 3–5 biologically independent samples, one-way ANNOVA, pairwise comparison, and exact p values are represented in the figure. Data are presented as mean ± standard deviation. (C) Chromatin accessibility at the Actg2 locus. ATAC-seq analysis reveals that chromatin accessibility at the Actg2 locus remains unchanged in the catalytically active T3H48-HHR knock-in allele as compared with the catalytically inactive one; position of the open chromatin peaks observed in Actg1^PTC1 mutant cells marked as in Fig. 1C. n = 3 biologically independent samples. Source data are available online for this figure.

upregulation of *Actg2*, and that this TA-like response of *Actg2* is not associated with increased chromatin accessibility of its locus. We also found that T3H48-HHR-catalyzed *Actg1* pre-mRNA cleavage is sufficient to knock down *Actg1* to levels similar as in *Actg1^PTC1* cells and to induce the strongest *Actg2* upregulation observed so far, and that without causing changes in *Actg2* chromatin accessibility. Altogether, these results provide additional evidence that RNA decay, and/or its products, rather than factors

associated with specific mRNA decay pathways, such as a PTC, is the key trigger for TA (Fig. 4).

So far, the nature of the *Actg1* RNA decay fragments that ultimately lead to *Actg2* upregulation remains elusive. It is expected that SMG6, the endonuclease involved in NMD, Cas13d, and T3H48-HHR recognize different cleavage sites in *Actg1* mRNA/ pre-mRNA; yet, the transcriptional upregulation of *Actg2* in all these experimental conditions may suggest somewhat converging

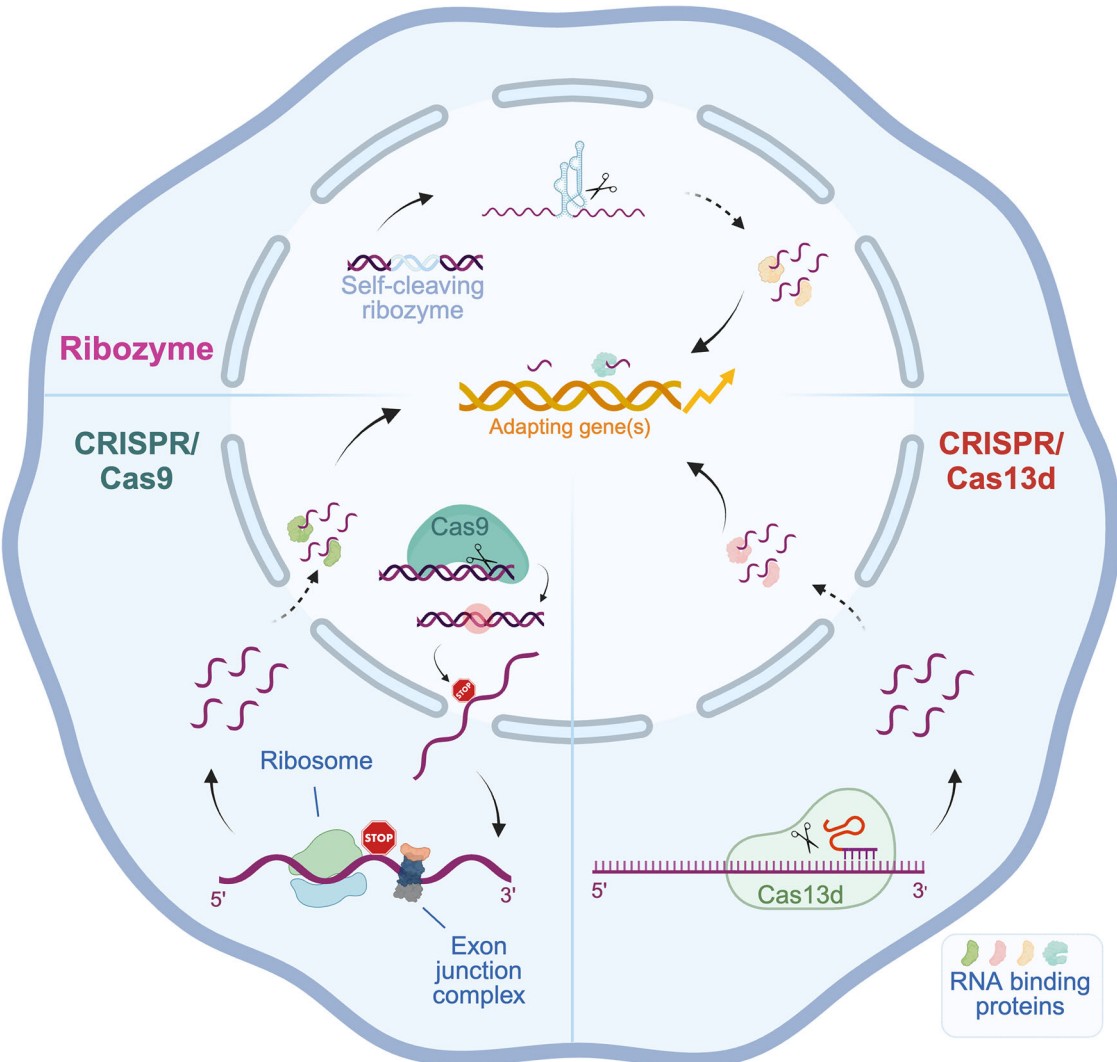

**Figure 4. Proposed model for TA/TA-like responses.**

Three different means of RNA destabilization, namely Cas9-induced mutations, Cas13d-mediated mRNA cleavage, and ribozyme-triggered pre-mRNA cleavage, lead to the transcriptional upregulation of the adapting gene(s). Cas9-induced frameshift mutations can result in PTC-containing mutant mRNAs that are recognized by the NMD machinery and degraded. Cas13d, an endoribonuclease, is guided to specific transcripts by gRNAs and initiates their decay. A self-cleaving ribozyme integrated into an intron of a gene of interest rapidly cleaves the pre-mRNA and silences the target gene. Through yet unknown mechanisms, (m)RNA fragments by themselves or in association with RNA-binding proteins translocate to the nucleus, if necessary, where they modulate gene expression. Created with BioRender.com.

RNA decay pathways and open up new avenues to explore the mechanisms underlying TA.

Another critical question is how RNA decay fragments modulate adapting gene expression. Historically, events that lead to TA, namely NMD or recognition and association of Upf3a with PTC-bearing mRNAs, were primarily linked with the cytoplasmic cellular compartment. Interestingly, some microRNAs have been reported to translocate from the cytoplasm to the nucleus with AGO2 and bind to specific promoter-associated non-coding RNAs, thereby leading to the release of transcriptional pausing and active elongation (Ohno et al, 2022). Furthermore, H3K4me3, a histone mark with increased abundance around the TSS of adapting genes during TA (El-Brolosy et al, 2019; Jiang et al, 2022), is required to switch Pol II from a paused to an elongating state (Wang et al,

2023). Together these findings suggest that transcriptional pausing regulated by small RNAs and histone modifications may be an integral part of the TA machinery. Nevertheless, it is also possible that the signal for TA originates in the nucleus. For example, it has been shown that aberrantly processed pre-mRNAs are degraded by the XRN2 exonuclease in mammalian nuclei (Davidson et al, 2012), likely yielding similar RNA fragments as those generated by cytoplasmic nucleases. The processing and decay of aberrant pre-mRNAs remains a scarcely studied phenomenon that requires broader attention, especially in the context of TA.

We have also reported that TA can play an unexpected role in obscuring the interpretation of gene perturbation phenotyping studies (El-Brolosy and Stainier, 2017; Jakutis and Stainier, 2021; Juvik et al, 2025). The data presented here suggest taking caution

not only in evaluating Cas9-mediated knockout studies, but also Cas13d-mediated knockdown studies. Although to date, TA in various model systems has been reported only as a consequence of mutant mRNA decay (Rossi et al, 2015; El-Brolosy et al, 2019; Ma et al, 2019; Ye et al, 2019; Serobyan et al, 2020; Welker et al, 2023) (for review see (Kontarakis and Stainier, 2020; Sztal and Stainier, 2020; Jakutis and Stainier, 2021)), our data indicate that other ways of cytoplasmic and nuclear RNA cleavage and decay can also lead to the transcriptional upregulation of paralogs. Therefore, we speculate that TA is a highly prevalent phenomenon that can be triggered by different modes of RNA decay and would like to further emphasize the importance of scrutinizing the genotype-phenotype relationship in future endeavors studying gene function.

# Methods

### Reagents and tools table

| Reagent/Resource | Reference or Source | Identifier or Catalog Number |
|---|---|---|
| **Experimental models** | | |
| NIH3T3 (*M. musculus*) | ATCC | CRL-1658 |
| **Recombinant DNA** | | |
| pSpCas9(BB)-2A-GFP | Addgene (a gift from Feng Zhang (Ran et al, 2013)) | 48138 |
| pX330-U6-Chimeric_BB-CBh-hSpCas9 | Addgene (a gift from Feng Zhang (Cong et al, 2013)) | 42230 |
| pUC19 | Addgene (a gift from Joachim Messing (Norrander et al, 1983)) | 50005 |
| pT3TS-RfxCas13d-HA | Addgene (a gift from Ariel Bazzini and Miguel Angel Moreno-Mateos (Kushawah et al, 2020)) | 141320 |
| pT3TS-RfxCas13d-NLS-HA | Addgene (a gift from Ariel Bazzini and Miguel Angel Moreno-Mateos (Kushawah et al, 2020)) | 141321 |
| **Antibodies** | | |
| Anti-gamma Actin antibody | Abcam | ab200046 |
| Anti-γ-Tubulin antibody | Sigma-Aldrich | T6557 |
| Goat anti-Rabbit IgG (H + L) Secondary Antibody, HRP | Invitrogen™ | 31460 |
| Goat Anti-Mouse IgG H&L (HRP) | Abcam | ab97023 |
| **Oligonucleotides and other sequence-based reagents** | | |
| Cas9 gRNAs | This study | Table EV1 |
| PCR primers | This study | Table EV1 |
| **Chemicals, Enzymes and other reagents** | | |
| DMEM, high glucose, GlutaMAX™ Supplement, pyruvate | Thermo Fisher Scientific | 31966021 |
| HyClone Iron-Supplemented Calf Serum | Cytiva | SH30072.03 |

| Reagent/Resource | Reference or Source | Identifier or Catalog Number |
|---|---|---|
| Penicillin-Streptomycin | Thermo Fisher Scientific | 15140122 |
| Puromycin | Sigma-Aldrich | P8833 |
| Lipofectamine™ 3000 Transfection Reagent | Thermo Fisher Scientific | L3000001 |
| TrypLE™ Express Enyzme | Thermo Fisher Scientific | 12604013 |
| PBS | Thermo Fisher Scientific | 10010023 |
| Lipofectamine™ RNAiMax Transfection Reagent | Thermo Fisher Scientific | 13778075 |
| Reduced Serum Medium | Thermo Fisher Scientific | 31985062 |
| BbsI | New England Biolabs | R0539S |
| T4 Polynucleotide Kinase | New England Biolabs | M0201S |
| T4 DNA ligase | New England Biolabs | M0202S |
| In-Fusion® Snap Assembly Master Mix | Takara Bio | 638948 |
| GeneJET Gel Extraction Kit | Thermo Scientific | K0691 |
| JumpStart™ REDTaq® ReadyMix™ Reaction Mix | Merck | P0982-100RXN |
| KAPA2G Fast HotStart PCR Kit | Roche | 08041202001 |
| MEGAshortscript™ T7 Transcription kit | Thermo Fisher Scientific | AM1354 |
| Lysis Buffer | Thermo Fisher Scientific | 46-6001 |
| 2-Mecaptoethanol | Thermo Fisher Scientific | 21985-023 |
| Poly(ethylene glycol), BioUltra, 8,000 | Sigma-Aldrich | 89510 |
| 5 M NaCl | Sigma-Aldrich | S6546 |
| 1 M Tris buffer, pH 8.0 | Millipore | 648314 |
| 0.5 M EDTA, pH 8.0 | PanReac AppliChem | A4892 |
| Tween® 20 | Sigma-Aldrich | P9416 |
| Carboxyl Magnetic Beads | Fisher Scientific | 09-981-123 |
| Ethanol | Fisher Scientific | 5054.4 |
| DNase I | New England Biolabs | M0303S |
| MAXIMA cDNA Synthesis Kit | Thermo Fisher Scientific | K1671 |
| DyNAmo ColorFlash SYBR Green PCR mix | Thermo Fisher Scientific | F-416 |
| RIPA buffer | Sigma-Aldrich | R0278 |
| 4–20% Criterion™ TGX™ Precast Midi Protein Gel | Bio-Rad Laboratories | 5671094 |
| 0.2 µm polyvinylidene difluoride (PVDF) membrane | Bio-Rad Laboratories | 1704157 |
| Clarity Western ECL Substrate | Bio-Rad Laboratories | 1705061 |
| Nextera DNA Sample Preparation Kit | Illumina | |
| MinElute PCR Purification Kit | QIAGEN | 28004 |

| Reagent/Resource | Reference or Source | Identifier or Catalog Number |
|---|---|---|
| **Software** | | |
| Nucleotide BLAST | National Center for Biotechnology Information | |
| Design & Analysis Software 2.7.0 | Thermo Fisher Scientific | |
| Trimmomatic v.0.33 | Bolger et al, 2014 | |
| STAR 2.4.2a | Dobin et al, 2013 | |
| Picard 1.136 | http://broadinstitute.github.io/picard/ | |
| MACS2 peak caller v.2.1.0 | | |
| Integrated Genomics Viewer (IGV) 2.3.52 | Robinson et al, 2011 | |
| bamCoverage | Ramirez et al, 2014 | |
| bigWigAverageOverBed | UCSC Genome Browser Utilities, http://hgdownload.cse.ucsc.edu/downloads.html | |
| genomecov | Quinlan and Hall, 2010 | |
| **Other** | | |
| BD FACSAria™ III Cell Sorter | BD Biosciences | |
| DynaMag-96 side skirted magnetic rack | Thermo Fisher Scientific | 12321D |
| DynaMag-2 magnetic rack | Thermo Fisher Scientific | 12321D |
| QuantStudio™ 7 Pro Real-Time PCR System | Thermo Fisher Scientific | A43185 |
| Trans-Blot® Turbo™ Transfer System | Bio-Rad Laboratories | |
| ChemiDoc MP Imaging System | Bio-Rad Laboratories | |
| MOXI Z Mini Automated Cell Counter Kit | Orflo | |
| NextSeq500 platform | Illumina | |

## Methods and protocols

### Cell culture

NIH3T3 cells purchased from ATCC were cultured in DMEM containing high glucose, GlutaMAX™, and pyruvate (Gibco™, 31966021) supplemented with 10% bovine calf serum (Cytiva, SH30072.03), 100 U/ml penicillin and streptomycin (Gibco™, 15140122) at 37 °C in a 5% $CO_2$ incubator. Cells were split at 80–90% confluence and kept below 20 passages.

### Generation of stable Actg1 mutant cell lines using CRISPR/Cas9

(1) To generate the $Actg1^{PTC}$ and $Actg1^{LD}$ mutant alleles, Cas9 gRNAs (Table EV1) were cloned into pSpCas9(BB)-2A-GFP plasmid (a gift from Feng Zhang; Addgene plasmid #48138), following a published protocol (Ran et al, 2013). Briefly, pSpCas9(BB)-2A-GFP plasmid was linearized by BbsI (NEB, R0539S) at 37 °C overnight and subsequently purified by gel electrophoresis and gel extraction with GeneJET Gel Extraction Kit (Thermo Fisher Scientific, K0691). Single-stranded oligonucleotides with Cas9 target sites (Table EV1) were phosphorylated and annealed by T4 Polynucleotide Kinase (New England Biolabs, M0201S) to create double-stranded DNA with compatible overhangs for ligation to the linearized plasmid with T4 DNA ligase (NEB, M0202S).

(2) Wild-type NIH3T3 cells were seeded at a density of 150,000 cells per well in a 6-well plate and cultured for 24 h or until 70% confluence at the time of transfection.

(3) To transfect cells in each well of a 6-well plate, 3.75 µl of Lipofectamine™ 3000 reagent and 2.5 µg of DNA were mixed with 125 µl of Opti-MEM™ separately before the diluted DNA was added to the diluted Lipofectamine™ 3000 at a 1:1 (v/v) ratio and incubated for 10–15 min at room temperature. 250 µl of the Lipofectamine™ 3000 and DNA complex was added to each well in a dropwise manner.

(4) 48 h after transfection, GFP positive cells were sorted into single cells in 96-well plates using a BD FACSAria™ III Cell Sorter and cultured for two weeks before genotyping the Actg1 locus with the primers listed in Table EV1. JumpStart™ REDTaq® ReadyMix™ Reaction Mix (Merck, P0982-100RXN) was used to genotype small indels in the third exon of Actg1 and KAPA2G Fast HotStart PCR Kit (Roche, 08041202001) for large deletions.

Actg1 is tetraploid in NIH3T3 cells (Leibiger et al, 2013). Based on genotyping and sequencing results, $Actg1^{PTC1}$ cells contain two different lesions, c.280_281insT and c.[280 A > G; 280_281insT], both leading to the same PTC. $Actg1^{PTC2}$ ($Actg1$-202:r.1096_1097insn[4036]) cells express an mRNA that lacks part of exon 5 and all of exon 6 but contains a 4036 bp insertion at $Actg1$-202:r.1096_1097 comprising a termination codon 66 bp 3' of the insertion site and forming a long 3'UTR. $Actg1^{LD}$ cells possess three different lesions: (1) complete absence of the gene, (2) absence of the gene except for the first 44 bases of the 5'UTR, and (3) absence of the gene except for the last 441 bases of the 3'UTR.

### Generation of stable Cas13d and Cas13d-NLS expressing cell lines

A safe harbor locus $Hipp11$ (Hippenmeyer et al, 2010) was selected for the targeted knock-in of Cas13d and Cas13d-NLS (Kushawah et al, 2020) in NIH3T3 cells.

(1) A gRNA targeting $Hipp11$ (Table EV1) was cloned into the Cas9-encoding pX330 vector (a gift from Feng Zhang; Addgene plasmid #42230), as previously described (Cong et al, 2013).

(2) Wild-type NIH3T3 cells were seeded in a 6-well plate and transfected with the pX330 plasmid containing $Hipp11$-targeting gRNA as well as a template plasmid containing Cas13d or Cas13d-NLS flanked by $Hipp11$ homology arms, eGFP, and PuroR cassette using Lipofectamine™ 3000 Transfection Reagent (Invitrogen™, L3000001), as described above.

(3) Transfected cells were selected with 8 µg/ml puromycin (Sigma-Aldrich, P8833) for three days.

(4) Then, cells expressing eGFP were sorted into single cells in 96-well plates using a BD FACSAria™ III Cell Sorter.

(5) 14 days after sorting, single clones were isolated and genotyped by PCR and sequencing. We grew 15 clones expressing Cas13d and 15 clones expressing Cas13d-NLS and after RT-qPCR analysis, selected four high expressors of each effector for the experiments.

### Cas13d gRNA assembly

The same rules for gRNA design and the same procedures for gRNA transfection apply to Cas13d in the cytoplasm and Cas13d-NLS in the nucleus. Therefore, for the sake of simplicity, Cas13d refers to both proteins in the Materials and Methods section hereafter.

(1) Cas13d target sites, located in exons or introns, were selected using an in-house generated gRNA efficiency calculator derived from a recent model (Wessels et al, 2020). gRNA candidates were screened with Nucleotide BLAST (National Center for Biotechnology Information) to eliminate those with potential off-target effects.

(2) Template for gRNA synthesis was assembled by annealing two oligonucleotides (Table EV1): a constant short oligonucleotide containing the T7 promoter and a variable long oligonucleotide containing the T7 promoter, a 23 bp target-specific sequence, and the scaffold sequence (Hruscha et al, 2013; Wessels et al, 2020). The three-step annealing process includes 5 min of incubation at 95 °C, a rapid cool down phase from 95 °C to 85 °C at −2 °C/s ramp rate, and a slow cool down phase from 85 °C to 25 °C at −0.1 °C/s ramp rate.

(3) Once the template was annealed, gRNAs were synthesized in vitro using a MEGAshortscript™ T7 Transcription kit (Thermo Fisher Scientific, AM1354) and directly used for transfection, or stored at −80 °C.

### Cas13d gRNA transfections for RT-qPCR experiments

(1) Cas13d-expressing NIH3T3 cells were seeded at a density of 6,000 cells per well in a 96-well plate and cultured for 24 h before transfection.

(2) gRNAs were transfected using Lipofectamine™ RNAiMax Transfection Reagent (Thermo Fisher Scientific, 13778075) with a modified protocol. Briefly, 1.5 μl of Lipofectamine™ RNAiMax Transfection Reagent or 150 ng of gRNA were diluted in 10.5 μl of Reduced Serum Medium (Gibco™, 31985062) separately before being mixed together; after 5 min of incubation at room temperature, 23 μl of the mixed reagents were added to each well in a 96-well plate.

(3) Samples were collected 14 h after transfection.

### Stable knock-in of a self-cleaving ribozyme in the third intron of Actg1

To trigger nuclear RNA destabilization, T3H48-HHR, a self-cleaving ribozyme (Zhong et al, 2020), was integrated into the Actg1 locus in wild-type NIH3T3 cells. As a negative control, catalytically inactive T3H48-HHR (or T3H48-iHHR), which harbors a single nucleotide substitution, was integrated into the same locus and compared with its catalytically active counterpart (or T3H48-aHHR).

(1) Cas9 gRNAs (Table EV1) targeting the third intron of Actg1 were cloned into pSpCas9(BB)-2A-GFP plasmid, as previously reported (Ran et al, 2013).

(2) A homologous recombination template, including a 257 bp left homology arm from Actg1, a 65 bp T3H48-HHR insulated by

CAAACAAACAAA (Wurmthaler et al, 2019) on both ends, and a 593 bp right homology arm from Actg1 were ligated to the pUC19 backbone by In-Fusion Cloning (Takara). The template was also flanked by the same Actg1 intron-targeting Cas9 gRNA sequences, which allowed it to be excised from the pUC19 plasmid backbone upon co-transfection with the gRNA-containing pSpCas9(BB)-2A-GFP plasmid.

(3) Wild-type NIH3T3 cells were seeded, transfected, and sorted as described above for generating stable Actg1 mutant cell lines using CRISPR/Cas9, except for one modification on the amount of transfected plasmids. In total, 1 μg of pSpCas9(BB)-2A-GFP plasmid containing the Actg1 intron-targeting gRNA and 3 μg of pUC19 plasmid containing the homologous recombination template were transfected into each well.

(4) To screen for homozygous T3H48-HHR integration, two pairs of primers, within or flanking the insulated T3H48-HHR sequence, were used to genotype multiple clones.

(5) In addition, full-length Actg1 mRNA was extracted using SPRI beads (detailed protocol can be found in the section "Gene expression analysis by RT-qPCR"), reverse transcribed using a MAXIMA cDNA Synthesis Kit (ThermoFisher Scientific, K1671), amplified by PCR, and analyzed by Sanger sequencing to make sure that the T3H48-HHR insertions did not interfere with RNA splicing.

In total, two Actg1$^{aHHR}$ cell lines and one Actg1$^{iHHR}$ cell line were established and further analyzed for gene expression. Notably, Actg1$^{aHHR1}$ cells and Actg1$^{aHHR2}$ cells exhibit different levels of Actg1 mRNA, likely due to different self-cleaving efficiencies of their respective T3H48-HHR modules. Yet, future work is required to understand the mechanisms underlying this difference.

### Gene expression analysis by RT-qPCR

(1) Wild-type NIH3T3 cells, as well as Actg1$^{PTC1}$, Actg1$^{PTC2}$, Actg1$^{LD}$, Actg1$^{aHHR}$, and Actg1$^{iHHR}$ mutant cells were seeded at a density of 150,000 cells per well in a 6-well plate and cultured for 24 h or until 70-90% confluence at the time of harvesting samples. Cas13d-expressing cells were seeded and transfected as mentioned above.

(2) Freshly prepared cell lysis solution, containing 1% 2-Mecaptoethanol (Gibco™, 21985-023) in Lysis Buffer (Invitrogen™, 46-6001), was added directly to the cell monolayer after removing the medium. Optimal volumes for cell lysis buffer were 100 μl per well for a 96-well plate and 300 μl per well for a 6-well plate. Cell lysates were collected after 10–15 min of incubation at room temperature and, if necessary, could be stored at −80 °C until RNA extraction.

(3) SPRI beads were prepared according to the published protocol (http://bit.ly/Home-Brew_SPRI_Beads). In short, 9 g of Poly(-ethylene glycol), BioUltra, 8000 (Sigma-Aldrich, 89510), 10 ml of 5 M NaCl (Sigma-Aldrich, S6546), 500 μl of 1 M Tris buffer, pH 8.0 (Millipore, 648314), and 100 μl of 0.5 M EDTA, pH 8.0 (PanReac AppliChem, A4892) were added to a 50 ml conical tube and adjusted to a total volume of 49 ml with distilled water. After mixing the solution, 27 μl of Tween® 20 (Sigma-Aldrich, P9416) and 1 ml of Carboxyl Magnetic Beads (Fisher Scientific, 09-981-123), pre-washed and resuspended in TE buffer, were added and mixed thoroughly.

(4) All reagents and procedures were at room temperature unless otherwise specified. Cell lysate was mixed thoroughly with two volumes of SPRI beads and incubated for 15 min, allowing nucleic acids to bind to the SPRI beads. Nucleic acid-bound SPRI beads were separated from the solution by a 15-min incubation on a strong magnet (Invitrogen, 12321D) and then rinsed three times with an equal volume of freshly prepared 80% ethanol. Air-dried beads were taken off the magnet and nucleic acids were eluted in 60 μl nuclease-free water for 15 min prior to another 15-min incubation on the magnet to remove the beads.

(5) DNase treatment was carried out with DNase I (NEB, M0303S) at 37 °C for 30 min in a total volume of 60 μl.

(6) Another round of RNA clean-up with SPRI beads was repeated as described above. RNA was eluted in a final volume of 10 μl.

(7) cDNA was synthesized using a MAXIMA cDNA Synthesis Kit (ThermoFisher Scientific, K1671), as per manufacturer's protocol.

(8) qPCR reactions were set up with DyNAmo ColorFlash SYBR Green PCR mix (Thermo Fisher Scientific, F-416) in a total volume of 10 μl, including 5 μl of 2x Mastermix, 1 μM forward primer, 1 μM reverse primer, cDNA, and nuclease-free water. A standard program was run on a QuantStudio™ 7 Pro Real-Time PCR System (ThermoFisher Scientific, A43185) and data analysis was performed with Design & Analysis Software 2.7.0 from ThermoFisher Scientific. Gene expression levels were normalized to *Hprt*, a housekeeping gene. Fold change was calculated using the $2^{-\Delta\Delta Ct}$ method: the housekeeping gene's mean Ct value was deducted from the Ct values of the analyzed gene to obtain the ΔCt. ΔΔCt was then calculated by subtracting the average ΔCt of the control samples from the ΔCts of the experimental condition samples.

### Western blot analysis

(1) Cas13d-expressing NIH3T3 cells were seeded and transfected in the same way as they were for RT-qPCR analysis (see previous section for more details).

(2) Cells were lysed in RIPA buffer (Sigma, R0278) and cell lysates from three identical wells in a 96-well plate were pooled to increase sample input.

(3) Proteins were separated by electrophoresis on a 4–20% Criterion™ TGX™ Precast Midi Protein Gel (Bio-Rad Laboratories, #5671094) and transferred onto a 0.2 μm polyvinylidene difluoride (PVDF) membrane (Bio-Rad Laboratories, #1704157) using the Trans-Blot® Turbo™ Transfer System (Bio-Rad Laboratories), following manufacturer's protocol.

(4) After 1 h of blocking in 5% non-fat milk at room temperature, the membrane was gently agitated in blocking buffer containing 1:1000 diluted primary antibodies at 4 °C overnight. Anti-gamma Actin antibody (Abcam, ab200046) was used for detecting ACTG1 and Anti-γ-Tubulin antibody (Sigma, T6557) was used for detecting gamma-tubulin, the loading control.

(5) The membrane was washed in TBST buffer three times prior to staining with 1:5000 diluted secondary antibodies in blocking buffer at room temperature for 1 h. Goat anti-Rabbit IgG (H + L) Secondary Antibody, HRP (Invitrogen™, #31460) or Goat Anti-Mouse IgG H&L (HRP) (Abcam, #ab97023) was used based on the origin of the primary antibodies.

(6) Chemiluminescence was generated with Clarity Western ECL Substrate (Bio-Rad Laboratories, #1705061) and detected by the ChemiDoc MP Imaging System (Bio-Rad Laboratories).

### Cas13d gRNA transfections for ATAC-seq analysis

Sequential Cas13d gRNA transfections were performed in samples analyzed by ATAC-seq.

(1) Cas13d-expressing cells were seeded at a density of 100,000 cells per well in a 6-well plate and cultured for 24 h before transfection.

(2) 30 μl of Lipofectamine™ RNAiMax (Thermo Fisher Scientific, 13778075) and 4 μg of gRNA were diluted in 150 μl of Reduced Serum Medium (Gibco™, 31985062) each, gently mixed together, incubated for 5 min at room temperature, and transfected into cells in each well.

(3) 48 h later, cells were split and seeded in a new 6-well plate, followed by another round of gRNA transfection on the same day.

(4) 72 h later, cells were harvested for ATAC-seq library preparation.

### ATAC-seq library preparation

(1) NIH3T3 cells, cultured in 6-well plates, were detached with TrypLE™ Express Enyzme (Gibco™, 12604013) and washed once with PBS (Gibco™, 10010023).

(2) 50,000 cells were counted with a MOXI Z Mini Automated Cell Counter Kit (Orflo) and used for ATAC library preparation using the Tn5 transposase from the Nextera DNA Sample Preparation Kit (Illumina). Briefly, cells were resuspended in 50 μl PBS and mixed with 25 μl of tagmentation DNA buffer, 2.5 μl of Tn5, 0.5 μl of 10% NP-40 and 22 μl of nuclease-free water. The combination of cells and Tn5 was incubated at 37 °C for 30 min with occasional snap mixing.

(3) After the transposase treatment, the mixture was incubated for 30 min at 50 °C with 500 mM EDTA, pH 8.0, for optimal recovery of the digested DNA fragments.

(4) 100 μl of 50 mM $MgCl_2$ was added to neutralize EDTA, and DNA fragments were purified using the MinElute PCR Purification Kit (QIAGEN, 28004).

(5) Amplification of the library together with indexing was performed as previously described (Buenrostro et al, 2013).

(6) Libraries were mixed in equimolar ratios and sequenced on a NextSeq500 platform using v2 chemistry.

### ATAC-seq data analysis

(1) Quality assessment of the samples was performed using FastQC (http://www.bioinformatics.babraham.ac.uk/projects/fastqc). Trimmomatic v.0.33 (Bolger et al, 2014) was employed to trim reads when their quality dropped below an average of Q20 within a five-nucleotide window. Only reads exceeding 30 nucleotides in length were retained for subsequent analyses.

(2) To standardize all samples to the same sequencing depth, 27 million reads per sample were randomly selected for further examination. These reads were aligned to the Ensembl mouse genome version mm10 (GRCm38) using STAR 2.4.2a (Dobin et al, 2013), with a focus on unique alignments to exclude ambiguously placed reads. Deduplication of reads was conducted using Picard 1.136 (http://broadinstitute.github.io/picard/) to eliminate PCR artifacts that could lead to multiple copies of the same original fragment.

(3) MACS2 peak caller v.2.1.0 was used for peak identification, with a minimum q value set to −1.5 and a false discovery rate of 0.01. To determine significant peaks, the data underwent manual inspection in the Integrated Genomics Viewer (IGV) 2.3.52 (Robinson et al, 2011). Peaks overlapping blacklisted regions (associated with known mis-assemblies and satellite repeats) from ENCODE were excluded. To compare peaks across samples, the resulting lists of significant peaks were merged to represent identical regions.

(4) After converting binary alignment map (BAM) files to bigWig format using deepTools bamCoverage (Ramirez et al, 2014), counts per unified peak per sample were computed with bigWigAverageOverBed (UCSC Genome Browser Utilities, http://hgdownload.cse.ucsc.edu/downloads.html). Raw counts for unified peaks were then normalized using DESeq2 for analysis (Anders and Huber, 2010). Spearman correlations were calculated to assess the reproducibility between samples in R.

(5) To facilitate a normalized display of samples in IGV, the original BAM files were adjusted for sequencing depth (based on the number of mapped deduplicated reads per sample) and noise level (reflecting the number of reads within peaks). This task was achieved through the computation of two factors applied to the original BAM files using bedtools genomecov (Quinlan and Hall, 2010), resulting in normalized bigWig files suitable for IGV.

## Data availability

The datasets produced in this study are available in the following databases: ATAC-Seq data: Gene Expression Omnibus GSE255121.

The source data of this paper are collected in the following database record: biostudies:S-SCDT-10_1038-S44319-025-00427-3.

## Peer review information

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

## Acknowledgements

We thank Kuan-Lun Hsu for his help with genotyping the *Actg1*[LD] allele, and Kuan-Lun Hsu and Mikhail Sharkov for their help with qPCR primer design. We also thank Carsten Kuenne for informatics support and Kikhi Khrievono for FACS support. This research was supported by funds from the Max Planck Society and awards from the European Research Council (ERC) under the European Union's research and innovation programs (AdG 694455-ZMOD and AdG 101021349-TAaGC) to DYRS. Gabrielius Jakutis was supported by a predoctoral fellowship from Boehringer Ingelheim Fonds. The funders had no role in study design, data collection and analysis, decision to publish, or preparation of the manuscript.

## Author contributions

**Lihan Xie**: Conceptualization; Data curation; Formal analysis; Validation; Investigation; Methodology; Writing—original draft; Project administration; Writing—review and editing. **Gabrielius Jakutis**: Conceptualization; Formal analysis; Investigation; Visualization; Methodology; Writing—original draft; Writing—review and editing. **Christopher M Dooley**: Conceptualization; Data curation; Formal analysis; Investigation; Methodology; Writing—original draft; Project administration; Writing—review and editing. **Stefan Guenther**: Data curation; Investigation. **Zacharias Kontarakis**: Conceptualization; Formal analysis; Methodology. **Sarah P Howard**: Validation; Investigation. **Thomas Juan**: Conceptualization; Resources; Formal analysis; Supervision; Methodology; Writing—original draft; Writing—review and editing. **Didier Y R Stainier**: Conceptualization; Formal analysis; Supervision; Funding acquisition;

Investigation; Writing—original draft; Project administration; Writing—review and editing.

Source data underlying figure panels in this paper may have individual authorship assigned. Where available, figure panel/source data authorship is listed in the following database record: biostudies:S-SCDT-10_1038-S44319-025-00427-3.

## Funding

## Disclosure and competing interests statement

The authors declare no competing interests.

# Expanded View Figures

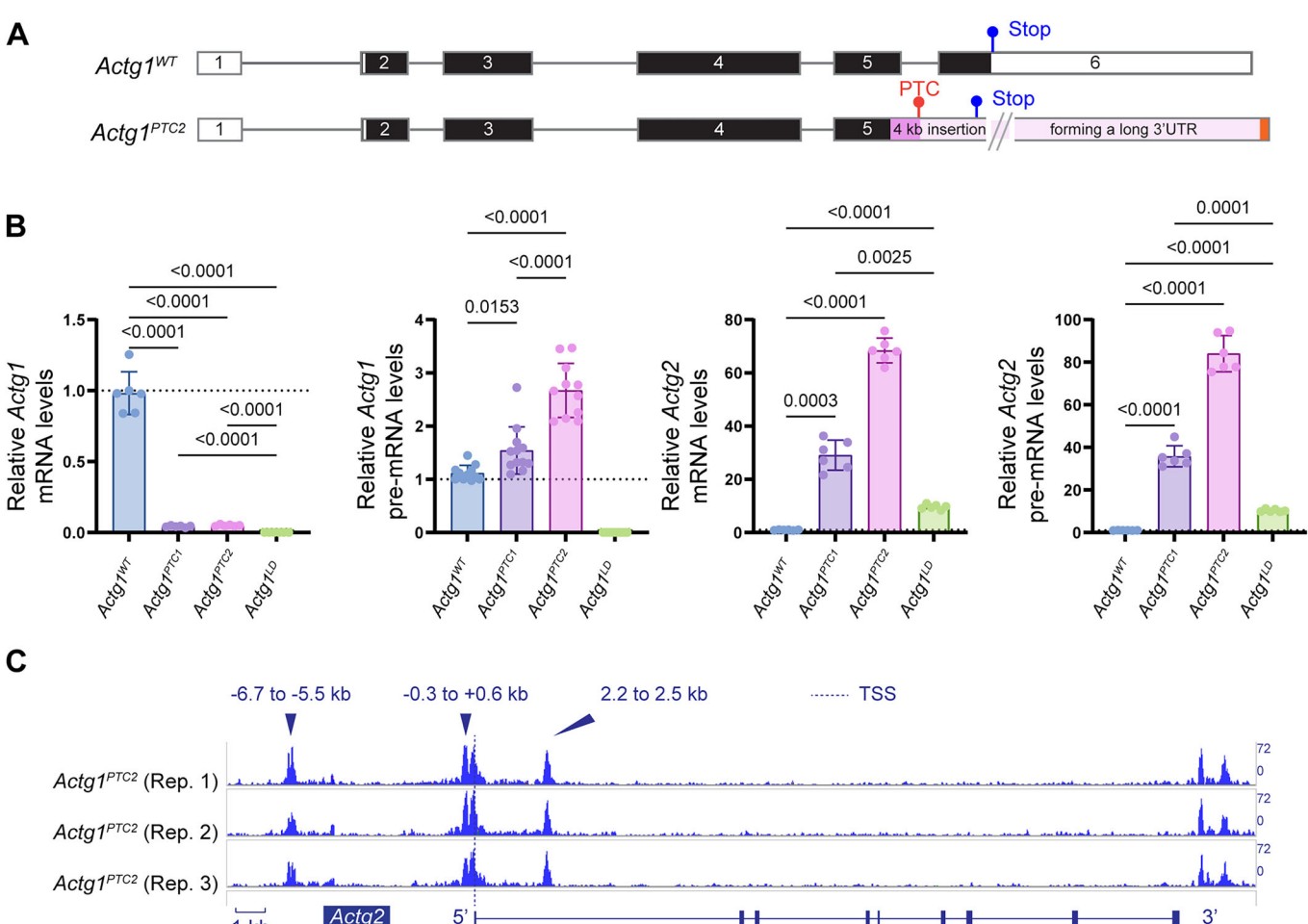

**Figure EV1. A second Cas9-induced *Actg1* mutation also leads to mutant mRNA decay and *Actg2* upregulation.**

(A) Schematic view of *Actg1*^WT^ and *Actg1*^PTC2^. Detailed information about the genotype of *Actg1*^PTC2^ cells can be found in the Materials and Methods section. (B) Relative mRNA and pre-mRNA levels of *Actg1* and *Actg2*. $n = 6$–12 biologically independent samples, one-way ANNOVA, pairwise comparison, and exact *p* values are represented in the figure. Data are presented as mean ± standard deviation. (C) Chromatin accessibility at the *Actg2* locus. ATAC-seq analysis reveals three open chromatin regions in the *Actg1*^PTC2^ allele located (1) in the 5′ intergenic region (i.e., −6.7 to −5.5 kb upstream of the transcription start site (TSS)), (2) around the TSS (i.e., −0.3 to +0.6 kb), and (3) in the first intron (i.e., 2.2 to 2.5 kb downstream of the TSS) of *Actg2*; $n = 3$ biologically independent samples.

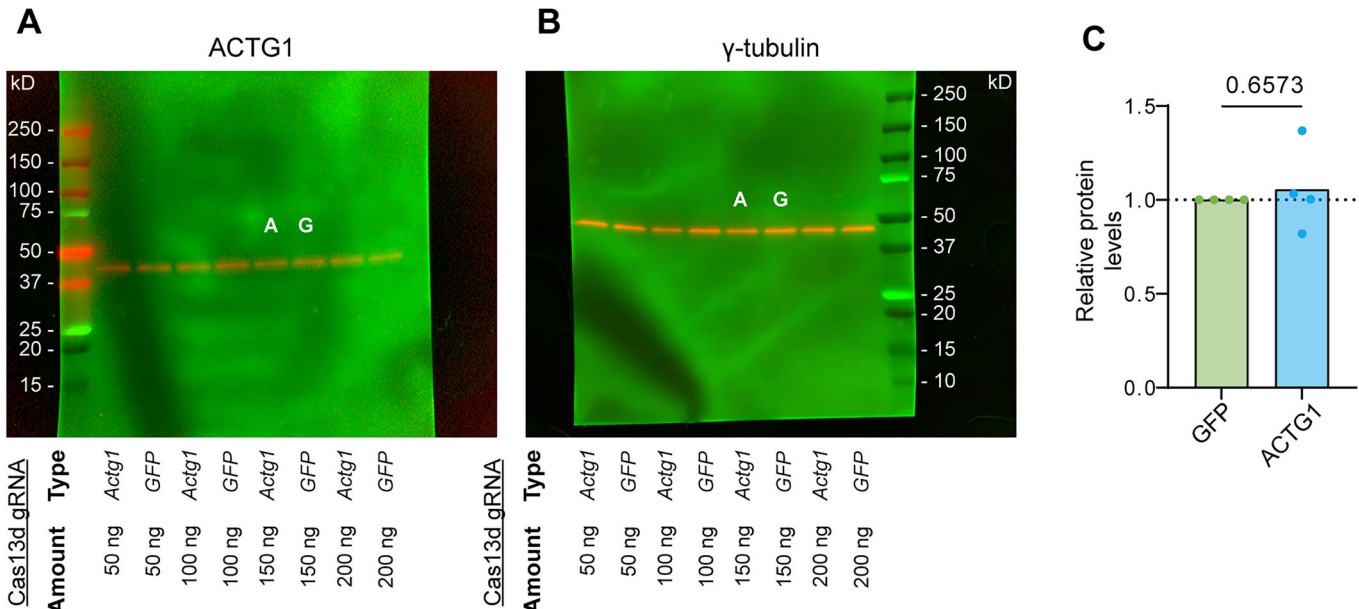

**Figure EV2. Cas13d-mediated *Actg1* mRNA cleavage does not lead to a substantial loss of ACTG1 protein.**

(A, B) Western blot analysis for ACTG1 (A) and γ-tubulin (B) from Cas13d-expressing cells treated with various amounts of *Actg1*- or *GFP*-targeting gRNAs for 14 h; 150 ng of *Actg1* and *GFP* gRNAs were used for the experiments shown in Fig. 2A, B. The molecular weight of ACTG1 is 41,793 Da, and that of γ-tubulin is 51,122 Da. Full uncropped views of blots shown in Fig. 2B; letters A and G on top of the bands in both blots refer to A – *Actg1* and G – *GFP*. (C) Quantification of ACTG1 Western blot bands following *Actg1* and *GFP* gRNA experiments; *n* = 4 biologically independent samples, unpaired t-test, and exact *p* values are represented in the figure.

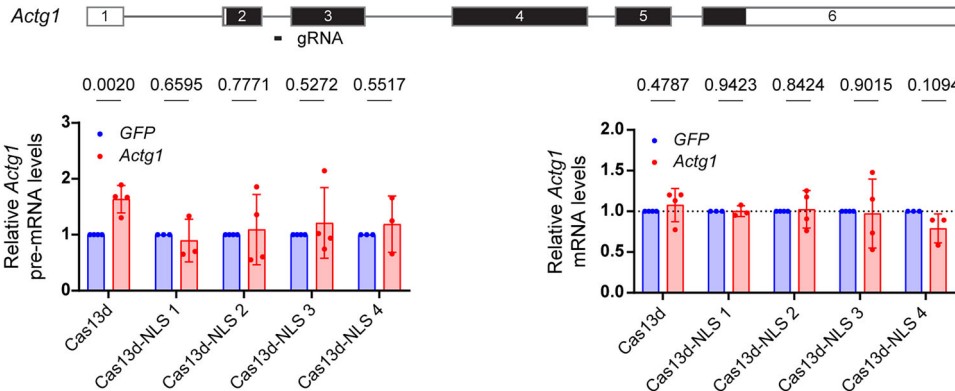

**Figure EV3.  Targeting *Actg1* pre-mRNA in Cas 13d-NLS cells.**

Top: position of Cas13d-NLS gRNA targeting *Actg1* intron 2. Bottom: *Actg1* pre-mRNA targeting does not lead to changes in *Actg1* pre-mRNA (left) or mRNA (right) levels in four independent Cas13d-NLS knock-in clones, compared with cytoplasmic Cas13d-expressing cells transfected with the intron 2 targeting gRNA. $n = 3$–4 biologically independent samples, unpaired t-test, and exact $p$ values are represented in the figure. Data are presented as mean ± standard deviation.

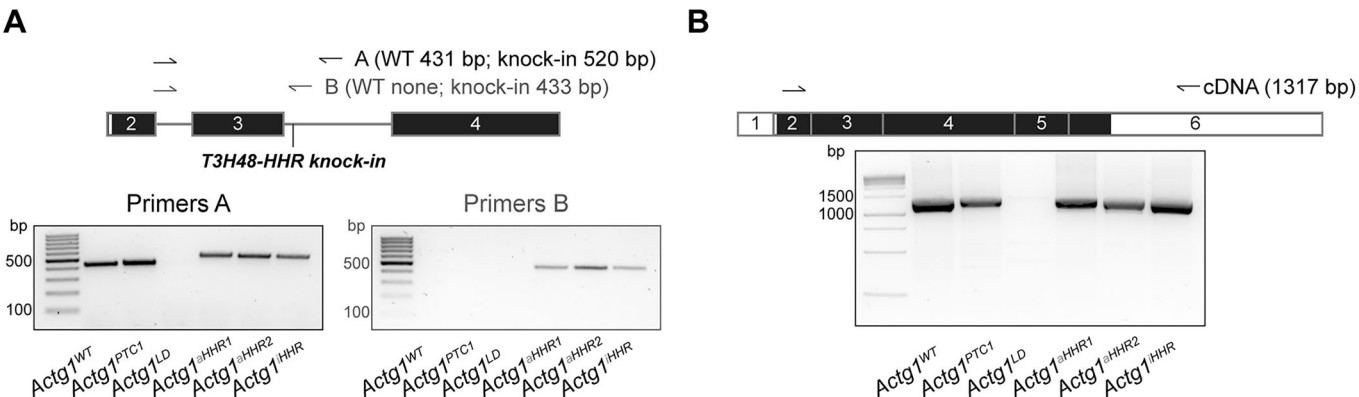

**Figure EV4. Validation of the T3H48-HHR knock-in cell lines.**

(A) Homozygous knock-in of the T3H48 ribozyme in intron 3 of *Actg1* as confirmed by two pairs of genotyping primers in *Actg1*[aHHR] and *Actg1*[iHHR] cells. Two independent clones were generated for *Actg1*[aHHR]. (B) *Actg1* cDNA profile in *Actg1*[aHHR] and *Actg1*[iHHR] cells is identical to that in wild-type cells, indicating no alternative splicing caused by the T3H48-HHR knock-in. RT-PCR reactions were run at saturation and therefore, *Actg1* downregulation was not observed in the PTC and T3H48-aHHR alleles in this experiment.

