## [Peer Review File · EMBO Reports]

Induction of a transcriptional adaptation response by RNA destabilization events

Lihan Xie, Gabrielius Jakutis, Christopher Dooley, Stefan Guenther, Zacharias Kontarakis, Sarah Howard, Thomas Juan, and Didier Stainier

Corresponding author(s): *Didier Stainier (Didier.Stainier@mpi-bn.mpg.de)* , *Thomas Juan (thomas.juan@mpi-bn.mpg.de)*

Review Timeline:

Submission Date:	14th Aug 24
Editorial Decision:	12th Sep 24
Revision Received:	24th Dec 24
Editorial Decision:	6th Feb 25
Revision Received:	3rd Mar 25
Accepted:	10th Mar 25

Editor: Esther Schnapp

Transaction Report:

Dear Didier,

Thank you for the submission of your manuscript to EMBO reports. We have now received the full set of referee reports that is pasted below.

As you will see, the referees acknowledge that the findings are potentially interesting. However, they also have a few suggestions for how the study could be improved and strengthened. I think all suggestions are good and should be addressed, but please let me know in case you disagree, and we can discuss the exact revision requirements further, also in a video chat, if you like.

I would thus like to invite you to revise your manuscript with the understanding that the referee concerns must be fully addressed and their suggestions taken on board. Please address all referee concerns in a complete point-by-point response. Acceptance of the manuscript will depend on a positive outcome of a second round of review. It is EMBO reports policy to allow a single round of major revision only and acceptance or rejection of the manuscript will therefore depend on the completeness of your responses included in the next, final version of the manuscript.

We realize that it is difficult to revise to a specific deadline. In the interest of protecting the conceptual advance provided by the work, we recommend a revision within 3 months (13th Dec 2024). Please discuss the revision progress ahead of this time with the editor if you require more time to complete the revisions.

You can either publish the study as a short report or as a full article. For short reports, the revised manuscript should not exceed 29,000 characters (including spaces but excluding materials & methods and references) and 5 main plus 5 expanded view figures. The results and discussion sections must further be combined, which will help to shorten the manuscript text by eliminating some redundancy that is inevitable when discussing the same experiments twice. For a normal article there are no length limitations, but it should have more than 5 main figures and the results and discussion sections must be separate. In both cases, the entire materials and methods must be included in the main manuscript file.

- 1) A data availability section providing access to data deposited in public databases is missing. If you have not deposited any data, please add a sentence to the data availability section that explains that.
- 2) Your manuscript contains statistics and error bars based on $n=2$. Please use scatter blots in these cases. No statistics should be calculated if $n=2$.

3) We replaced Supplementary Information with Expanded View (EV) Figures and Tables that are collapsible/expandable online. A maximum of 5 EV Figures can be typeset. EV Figures should be cited as 'Figure EV1, Figure EV2' etc... in the text and their respective legends should be included in the main text after the legends of regular figures.

5) a complete author checklist, which you can download from our author guidelines . Please insert information in the checklist that is also reflected in the manuscript. The completed author checklist will also be part of the RPF.

6) Please note that all corresponding authors are required to supply an ORCID ID for their name upon submission of a revised manuscript (. Please find instructions on how to link your ORCID ID to your account in our manuscript tracking system in our Author guidelines

- the name of the statistical test used to generate error bars and P values,
- the number (n) of independent experiments (please specify technical or biological replicates) underlying each data point,
- the nature of the bars and error bars (s.d., s.e.m.),
- If the data are obtained from n {less than or equal to} 2, use scatter blots showing the individual data points.

12) All Materials and Methods need to be described in the main text using our 'Structured Methods' format, which is required for all research articles. According to this format, the Methods section includes a Reagents and Tools Table (listing key reagents, experimental models, software and relevant equipment and including their sources and relevant identifiers) and a Methods and Protocols section describing the methods using a step-by-step protocol format. The aim is to facilitate adoption of the methodologies across labs. More information on how to adhere to this format as well as a downloadable template (.docx) for the Reagents and Tools Table can be found in our author guidelines:
<https://www.embopress.org/page/journal/14693178/authorguide#structuredmethods>.

An example of a Method paper with Structured Methods can be found here: <https://www.embopress.org/doi/full/10.1038/s44320-024-00037-6#sec-4>.

As part of the EMBO publication's Transparent Editorial Process, EMBO reports publishes online a Review Process File (RPF)

to accompany accepted manuscripts. This File will be published in conjunction with your paper and will include the referee reports, your point-by-point response and all pertinent correspondence relating to the manuscript.

I look forward to seeing a revised form of your manuscript when it is ready.

Referee #1:

Xie et al. examined the underlying mechanism of "transcriptional adaptation (TA)," an intriguing phenomenon in which compensatory genes are upregulated in expression in response to mRNA-destabilizing mutations. To do this, the authors generated several mouse NIH-3T3 lines that modified the *Actg1* gene or its expression. Their findings suggest that RNA degradation of *Actg1* mRNA is sufficient to trigger TA; i.e., upregulate the expression of its paralog *Actg2*.

Overall, this manuscript is well designed and provides results of great interest to the field. However, there are some concerns:

1. PTC specificity. The evidence that PTCs trigger the TA response in mammalian cells is limited. In this manuscript, only a single PTC is tested for its ability to elicit TA (Fig. 1A). At least one additional PTC should be tested. Optimally, different types of PTC-generating mutations (e.g., nonsense and frameshift mutations) should be examined.
2. Cas13d- vs. siRNA-mediated mRNA cleavage. The authors' finding that the TA response is triggered by Cas13d-mediated cytoplasmic mRNA cleavage (Fig. 2) is surprising given that a fundamental feature of the TA mechanism in zebrafish is that it is NOT triggered by siRNA-mediated knockdown (as demonstrated in the authors' 2019 Nature paper as well as the Ma et al. 2019 Nature paper). One possibility is that Cas13d-mediated mRNA decay and RNAi-mediated mRNA decay act by slightly different mechanisms and thus one triggers TA and the other does not. Another possibility is siRNAs do trigger TA in mammalian cells. The authors should begin to address this by determining whether siRNAs that knockdown *Actg1* trigger TA or not. Whether the authors find that RNAi elicits TA in mammalian cells or not, this is fundamental information. It is critical to know whether RNAi resistance is a conserved feature of the TA mechanism.

Minor concerns:

- (1) Apparent false claim. The authors claim in the first sentence of the R&D section that their previous 2019 paper did studies with NIH-3T3 cells containing a nonsense-mutant *Actg1* gene. I could find no such studies; only studies with *Actg1*-mutant MEFs and mESCs. This is important, as the present MS is on NIH-3T3 cells.
- (2) Is the TA-inducing signal nuclear? The authors found that nuclear mRNA decay (ribozyme-mediated pre-mRNA cleavage) triggers a much stronger TA response than cytoplasmic mRNA decay (Cas13d-mediated mRNA cleavage (compare Fig. 2 with Fig. 3). This raises the possibility the primary mechanisms responsible for TA originates in the nucleus. The authors should discuss this possibility.

Referee #2:

Subject: "Induction of a transcriptional adaptation response by RNA destabilization events."

This manuscript presents valuable insights for genetic compensation and transcription adaptation research, but several major concerns regarding the experimental design and scientific robustness need to be addressed. Below are some key suggestions to enhance the overall quality and strengthen the experimental framework of the study.

Overall, I believe that if these concerns are adequately addressed, the manuscript has the potential to make a significant contribution to EMBO Reports. I acknowledge and appreciate the considerable effort put forth by the authors in conducting this research, and I look forward to seeing the final version of the manuscript.

1. Figure 1A: Authors have successfully recapitulated their own results (El-Brolosy et al., 2019) by generating Cas9 mutants in NIH3T3 cell lines (a PTC in its third exon (denoted as *Actg1*PTC) as well as a new RNA-less allele (denoted as *Actg1*FLD) (Fig.

1A). The authors hypothesize that genetic compensation or transcriptional adaptation (TA) is potentially regulated by nonsense-mediated decay (NMD). However, it is unclear why the authors chose to target only the 3rd exon for introducing a premature termination codon (PTC) in the *Actg-1* gene. A more classical approach would involve generating a traditional PTC NMD model and comparing both to assess which system drives a stronger TA response. This would provide a direct comparison of different EJC-dependent and independent NMD models' contribution to TA response.

2. Figure 1C: Authors show that in ATAC seq experiment, chromatin remodeling near 5'end of *Actg2* gene suggesting transcriptional activation near transcription start site and in promoter region. It would be nice, if authors further compare these results with model proposed by Ma et al. (2019). "The genetic compensation response (GCR) is accompanied by an enhancement of histone H3 Lys4 trimethylation (H3K4me3) at the transcription start site regions of the compensatory genes.

3. Figure 2: The authors demonstrate CRISPR-Cas13d-mediated genetic compensation in mouse NIH3T3 cell lines using Cas13d system from Kushawah et al. (2020). However, Kushawah et al. used zebrafish codon-optimized Cas13d plasmids. Therefore, the authors should either perform this study in zebrafish embryos using the recommended Cas13d system or, if they intend to replicate the Figure 1 results in NIH3T3 cell lines, they could use the plasmid resources described by S. Konermann (Cell, 2018) or some other cas13d mediated studies in cell lines.

4. Figure 2: Using Cas13d from Kushawah et al. (2020), in NIH3T3 cell lines, authors demonstrate knockdown of *Actg1* gene and weak TA for *Actg-2* (2-fold: Cas13d vs. 30-fold with Cas9 mutation, Figure 1). This could be due to moderate knockdown of *Actg1* gene, Authors should have used multiple gRNAs (2-3) targeting *Actg-1* (as they also did for *Ctnna1* and *Nckap1* mRNA) to induce strong knockdown effect or they may consider following the approach outlined by S. Konermann (Cell, 2018).

5. Figure 2A: It would be worthwhile to target *Actg-1* using multiple independent gRNAs, both individually and in combination, to assess the effect of varying *Actg-1* knockdown efficiencies on *Actg-2* transcriptional adaptation (TA). This approach could also demonstrate that the TA response can be modulated or fine-tuned based on knockdown strength.

6. Figure 2E: ATAC seq in Cas13d mediated *actg-1* knockdown cell (Figure 2E), authors don't see any change in chromatin accessibility near *Actg-2* promoter region or TSS region. Authors should test the effect of strong Cas13d mediated *actg-1* knockdown, as previously suggested to induce strong transcription adaptation. Once authors have strong evidence to show this, then only authors should conclude "Cas13d-mediated TA-like response does not require a significant extent of chromatin remodeling."

7. Figure 2 expanded view: I would like that authors should follow the approach described by S. Konermann (Cell, 2018) or other cell-based studies to test the effectiveness of Cas13d-NLS-mediated nuclear RNA depletion in mouse NIH3T3 cell lines rather than method used by Kushawah et al. (2020), which was tested in zebrafish embryos.

8. Figure 3: Using Hammerhead ribozyme (HHRs) against *Actg-1* gene, authors show even stronger TA response for *Actg-2* than Cas9 and Cas13d mediated TA. It would be nice if authors perform ATAC seq in this background to check the change in chromatin accessibility near *Actg-2* promoter region or TSS region.

9. Figure 4: The authors need to provide at least some evidence supporting the involvement of the nonsense-mediated decay (NMD) machinery in mediating this transcriptional adaptation (TA) model for gene compensation. One approach could be to target key NMD factors, such as UPF1, as mentioned in the introduction, or to explore the UPF3a competition model. These factors could be directly targeted either through Cas9-based genome editing or by depleting their transcripts using Cas13d.

Minor comments:

Expanded view Figure 1: Authors mentioned that they don't see any difference in the levels of control and ACTG-1 proteins in figure1EV A, while to rule out any doubt, authors could crop the gels to highlight only bands of ACTG1 and respective Y-tubulin bands as shown in figure (fig. 2B).

Referee #3:

This is an interesting submission that further characterizes the mechanism triggering "transcriptional adaptation" (TA). There are two theories put forth: one by the authors' lab (Stainier's), and the other by the Peng lab. This submission further substantiates Stainier's theory that, rather than UPF3a functioning in TA as proposed by Peng, RNA degradation (presumably via RNA decay products) formed either in the cytoplasm or, as shown here, in the nucleus, upregulate not only a paralogous gene(s) but the gene from which the RNA decay products derive. Other new findings include assays for open chromatin: open chromatin for the gene upregulated by TA is evident for TA due to a PTC-containing mRNA that is degraded by NMD but not an mRNA degraded

by CRISPR-Cas9 (the significance of which remains unclear - an explanation/comment from the authors would be helpful). This submission moves the field forward in a creative way. It also raises concerns about interpreting CRISPR-Cas9 gene KO data, which is useful for the scientific community to know.

Comments to be addressed by the authors include the following.

Specific Comments

Page 5, first paragraph. The meaning of "lacking a normal stop codon" requires clarification. Do the authors mean "containing a PTC"?

Page 5, first paragraph. No where is "FLD" defined. It must not mean "full-length deletion" since in Actg1FLD cells, there were three different lesions, presumably because there are three different alleles for this gene (?), and one lesion leaves behind the last 441 bases of the 3'UTR while another leaves behind 44 bases of the 5'UTR.

Page 5, first paragraph. Is RNA produced from the "last 441 bases of the 3'UTR"? This seems to be an important issue to resolve, as it is for the allele leaving behind the first 44 bases of the 5'UTR, which one would think would produce RNA. The authors want to claim that RNA production is required for TA. Thus, they should better characterize, or at the very least make a comment about, the possibility of Actg1 RNA production in the Actg1FLD cells. This reviewer realizes that no mRNA was detected in the steady-state using their RT-qPCR assay for mRNA. However, this may not mean that RNA wasn't produced and subsequently degraded in a way that would not be detected by their RT-qPCR assay.

Page 6, first paragraph. Is TA to the same extent when there is cytoplasmic mRNA degradation without a genomic lesion and when there is a PTC? It would be nice to have a quantitative comparison of Figures 1, 2 and 3. This is done in Figure 3, but without normalization to a cellular RNA (see next comment).

Figures 1,2 and 3. For the RT-qPCR to be quantitative, the authors need to normalize mRNA and pre-mRNA levels to a cellular RNA. Apparently, this hasn't been done, but it should be.

Page 7, first paragraph. Please change "TA or a TA-like response" to "TA vs. a TA-like response".

Page 8, last paragraph. Interestingly, the extent to which Actg1 pre-mRNA was cleaved by the HHR ribozyme correlated with "Actg2 upregulation". Please specify in the text how cleavage efficiency was assayed.

Max Planck Institute
for Heart and Lung Research
W.G. Kerckhoff Institute

Max Planck Institute for Heart and Lung Research
Ludwigstrasse 43 · D-61231 Bad Nauheim

Editor comments:

1. We replaced *Supplementary Information* with *Expanded View (EV) Figures and Tables* that are collapsible/expandable online. A maximum of 5 EV Figures can be typeset. EV Figures should be cited as 'Figure EV1, Figure EV2" etc... in the text and their respective legends should be included in the main text after the legends of regular figures.

We have adopted the new terminology and renamed the supplementary information as expanded view (EV) figures and/or tables in the manuscript.

- the name of the statistical test used to generate error bars and P values,
- the number (n) of independent experiments (please specify technical or biological replicates) underlying each data point,
- the nature of the bars and error bars (s.d., s.e.m.),
- If the data are obtained from n {less than or equal to} 2, use scatter blots showing the individual data points.

We have now ensured that all of the above mentioned points were specified in in each figure legend in the manuscript.

3. All Materials and Methods need to be described in the main text using our 'Structured Methods' format, which is required for all research articles. According to this format, the Methods section includes a Reagents and Tools Table (listing key reagents, experimental models, software and relevant equipment and including their sources and relevant identifiers) and a Methods and Protocols section describing the methods using a step-by-step protocol format. The aim is to facilitate adoption of the methodologies across labs.

We have now adopted the 'Structured Methods' format and rewrote the Materials and Methods section accordingly.

Reviewer #1:

Xie et al. examined the underlying mechanism of "transcriptional adaptation (TA)," an intriguing phenomenon in which compensatory genes are upregulated in expression in response to mRNA-destabilizing mutations. To do this, the authors generated several mouse NIH-3T3 lines that modified the Actg1 gene or its expression. Their findings suggest that RNA degradation of Actg1 mRNA is sufficient to trigger TA; i.e., upregulate the expression of its paralog Actg2.

Overall, this manuscript is well designed and provides results of great interest to the field.

We thank the reviewer for their recognition of the importance of this work.

1. *PTC specificity. The evidence that PTCs trigger the TA response in mammalian cells is limited. In this manuscript, only a single PTC is tested for its ability to elicit TA (Fig. 1A). At least one additional PTC should be tested. Optimally, different types of PTC-generating mutations (e.g., nonsense and frameshift mutations) should be examined.*

As suggested by the reviewer, we have now generated and analyzed an additional Actg1 mRNA decay-displaying allele (lines 93–95 and EV Figure 1); this allele behaves just like the original Actg1^{PTC} allele.

2. *Cas13d- vs. siRNA-mediated mRNA cleavage. The authors' finding that the TA response is triggered by Cas13d-mediated cytoplasmic mRNA cleavage (Fig. 2) is surprising given that a fundamental feature of the TA mechanism in zebrafish is that it is NOT triggered by siRNA-mediated knockdown (as demonstrated in the authors' 2019 Nature paper as well as the Ma et al. 2019 Nature paper). One possibility is that Cas13d-mediated mRNA decay and RNAi-mediated mRNA decay act by slightly different mechanisms and thus one triggers TA and the other does not. Another possibility is siRNAs do trigger TA in mammalian cells. The authors should begin to address this by determining whether siRNAs that knockdown Actg1 trigger TA or not. Whether the authors find that RNAi elicits TA in mammalian cells or not, this is fundamental information. It is critical to know whether RNAi resistance is a conserved feature of the TA mechanism*

We thank the reviewer for this important question. First, we would like to clarify that in our 2019 *Nature* paper as well as in the Ma et al. 2019 *Nature* paper, there is no siRNA work that demonstrates the ability or inability of RNAi to trigger TA. Instead, in our 2019 *Nature* paper, we used RNAi to knock down several factors of the NMD pathway to test their requirement for TA. We do however fully agree that determining whether siRNA-mediated knockdown can also lead to TA, or a TA-like response, is a fundamentally important question, and we are actively pursuing it. Addressing this question thoroughly will require the investigation of multiple models/genes, as well as the use of both siRNAs and shRNAs, i.e., a big project in itself that falls outside the scope of this manuscript.

3. Apparent false claim. The authors claim in the first sentence of the R&D section that their previous 2019 paper did studies with NIH-3T3 cells containing a nonsense-mutant Actg1 gene. I could find no such studies; only studies with Actg1-mutant MEFs and mESCs. This is important, as the present MS is on NIH-3T3 cells.

We would like to clarify that our 2019 *Nature* paper indeed included experiments using *Actg1* KO NIH-3T3 cells, which are immortalized MEFs (Todaro & Green, 1963). This information is included in the 2019 *Nature* paper Materials and methods section: “Repeated passage of EFs resulted in establishment of immortalized cell lines (3T3 cells) from both *RelA*^{+/+} and *RelA*^{-/-} EFs” (Beg & Baltimore, 1996). And none of the experiments in our 2019 *Nature* paper used *Actg1* KO mESCs; we did however use *Actb* KO mESCs.

4. Is the TA-inducing signal nuclear? The authors found that nuclear mRNA decay (ribozyme-mediated pre-mRNA cleavage) triggers a much stronger TA response than cytoplasmic mRNA decay (Cas13d-mediated mRNA cleavage (compare Fig. 2 with Fig. 3). This raises the possibility the primary mechanisms responsible for TA originates in the nucleus. The authors should discuss this possibility.

Thank you for this question. As suggested by the reviewer, we have now expanded the Results and Discussion section (lines 258–261) to address the possibility that TA primarily originates in the nucleus.

Reviewer #2:

This manuscript presents valuable insights for genetic compensation and transcription adaptation research, but several major concerns regarding the experimental design and scientific robustness need to be addressed. Below are some key suggestions to enhance the overall quality and strengthen the experimental framework of the study.

Overall, I believe that if these concerns are adequately addressed, the manuscript has the potential to make a significant contribution to EMBO Reports. I acknowledge and appreciate the considerable effort put forth by the authors in conducting this research, and I look forward to seeing the final version of the manuscript.

We thank the reviewer for their recognition of the importance of this work.

1. *Figure 1A: Authors have successfully recapitulated their own results (El-Brolosy et al., 2019) by generating Cas9 mutants in NIH3T3 cell lines (a PTC in its third exon (denoted as Actg1PTC) as well as a new RNA-less allele (denoted as Actg1FLD) (Fig. 1A). The authors hypothesize that genetic compensation or transcriptional adaptation (TA) is potentially regulated by nonsense-mediated decay (NMD). However, it is unclear why the authors chose to target only the 3rd exon for introducing a premature termination codon (PTC) in the Actg-1 gene. A more classical approach would involve generating a traditional PTC NMD model and comparing both to assess which system drives a stronger TA response. This would provide a direct comparison of different EJC-dependent and independent NMD models' contribution to TA response.*

We are not sure what the reviewer refers to by a 'traditional PTC NMD model'. The PTC generated in the third exon of *Actg1* complies with all NMD rules (Lindeboom et al., 2016) and is predicted to trigger mRNA decay via the EJC-dependent NMD pathway. We have generated and analyzed an additional *Actg1* mRNA decay-displaying allele and found that it behaves just like the original *Actg1*^{PTC} allele. We fully agree with the reviewer that testing several NMD models (e.g., different PTCs), as well as NGD and NSD models, will be very interesting as they might have different effects on the adapting gene(s) expression and lead to important mechanistic insights about TA.

2. *Figure 1C: Authors show that in ATAC seq experiment, chromatin remodeling near 5'end of Actg2 gene suggesting transcriptional activation near transcription start site and in promoter region. It would be nice, if authors further compare these results with model proposed by Ma et al. (2019). "The genetic compensation response (GCR) is accompanied by an enhancement of histone H3 Lys4 trimethylation (H3K4me3) at the transcription start site regions of the compensatory genes.*

In our previous paper (El-Brolosy et al. (2019)), we also observed an increase in chromatin accessibility near the transcription start site (TSS) of *Fermt1* (the adapting gene for *Fermt2*) in *Fermt2* knockout mouse kidney fibroblasts (Extended Data Figure 3c). This change was accompanied by an increase in WDR5 occupancy (Figure 4c) and H3K4me3 deposition (Figure 4d) around the TSS of *Fermt1*. Similarly, and as mentioned by the reviewer, Ma et al. (2019) showed an enrichment of H3K4me3 at the TSS of the compensatory genes in their models. Thus, our ATAC-seq results for the *Actg1* TA model agree with these previous results in suggesting that chromatin remodeling is a common feature of TA, and we have now clarified this point (and referenced the Ma et al. (2019) data) in the manuscript. However, future studies will be required to dissect the causal relationship between TA and such epigenetic changes, a very important question indeed.

3. *Figure 2: The authors demonstrate CRISPR-Cas13d-mediated genetic compensation in mouse NIH3T3 cell lines using Cas13d system from Kushawah et al. (2020). However, Kushawah et al. used zebrafish codon-optimized Cas13d plasmids. Therefore, the authors should either perform this study in zebrafish embryos using the recommended Cas13d system or, if they intend to replicate the Figure 1 results in NIH3T3 cell lines, they could use the plasmid resources described by S. Konermann (Cell, 2018) or some other cas13d mediated studies in cell lines.*

We would like to point out that Kushawah et al. (2020) used human codon-optimized Cas13d and NLS-Cas13d-NLS effectors, which were initially described by Konermann et al. (2018), and which we used in our study. The reference has now been fixed (line 125).

4. *Figure 2: Using Cas13d from Kushawah et al. (2020), in NIH3T3 cell lines, authors demonstrate knockdown of Actg1 gene and weak TA for Actg-2 (2-fold: Cas13d vs. 30-fold with Cas9 mutation, Figure 1). This could be due to moderate knockdown of Actg1 gene, Authors should have used multiple gRNAs (2-3) targeting Actg-1 (as they also did for Ctnna1 and Nckap1 mRNA) to induce strong knockdown effect or they may consider following the approach outlined by S. Konermann (Cell, 2018).*

Using multiple independent gRNAs to knock down *Actg1* mRNA is unfortunately not feasible due to the high sequence similarity between *Actg1* and *Actb*. When testing multiple *Actg1*-targeting guides, we indeed observed a significant downregulation of *Actb*, which itself leads to *Actg2* upregulation.

5. *Figure 2A: It would be worthwhile to target Actg-1 using multiple independent gRNAs, both individually and in combination, to assess the effect of varying Actg-1 knockdown efficiencies on Actg-2 transcriptional adaptation (TA). This approach could also demonstrate that the TA response can be modulated or fine-tuned based on knockdown strength.*

We indeed tried this approach when optimizing the experimental setup. As mentioned above, using multiple independent gRNAs to knock down *Actg1* mRNA is unfortunately not feasible due to the high sequence similarity between *Actg1* and *Actb*. The single guide used in our manuscript corresponds to the only 23-nucleotide sequence within *Actg1* that is unique and does not also target *Actb*. When testing multiple guides, we always observed significant downregulation of *Actb*.

6. *Figure 2E: ATAC seq in Cas13d mediated actg-1 knockdown cell (Figure 2E), authors don't see any change in chromatin accessibility near Actg-2 promoter region or TSS region. Authors should test the effect of strong Cas13d mediated actg-1 knockdown, as previously suggested to induce strong transcription adaptation. Once authors have strong evidence to show this, then only authors should conclude "Cas13d-mediated TA-like response does not require a significant extent of chromatin remodeling.*

As mentioned above, it is not feasible to induce a 'stronger' Cas13d-mediated *Actg1* downregulation using a multiple guide approach because of the high sequence similarity between *Actg1* and *Actb*. However, the new ATAC-seq data on the T3H48-HHR ribozyme line indicate that chromatin remodeling is not required for a TA-like response.

7. *Figure 2 expanded view: I would like that authors should follow the approach described by S. Konermann (Cell, 2018) or other cell-based studies to test the effectiveness of Cas13d-NLS-mediated nuclear RNA depletion in mouse NIH3T3 cell lines rather than method used by Kushawah et al. (2020), which was tested in zebrafish embryos.*

In fact, in our study, we did follow the workflow outlined in Figure 4A of S. Konermann et al. (2018) and measured gene expression by qPCR.

8. *Figure 3: Using Hammerhead ribozyme (HHRs) against Actg-1 gene, authors show even stronger TA response for Actg-2 than Cas9 and Cas13d mediated TA. It would be nice if authors perform ATAC seq in this background to check the change in chromatin accessibility near Actg-2 promoter region or TSS region.*

As suggested by the reviewer, we have now performed an ATAC-seq experiment in the T3H48-HHR knock-in cells. The data show that chromatin accessibility at the *Actg2* locus remains unchanged in the catalytically active T3H48-HHR knock-in allele as compared with the catalytically inactive one, and they are described in detail in the Results section (lines 226–232) and Figure 3C.

9. *Figure 4: The authors need to provide at least some evidence supporting the involvement of the nonsense-mediated decay (NMD) machinery in mediating this transcriptional adaptation (TA) model for gene compensation. One approach could be to target key NMD factors, such as UPF1, as mentioned in the introduction, or to explore the UPF3a competition model. These factors could be directly targeted either through Cas9-based genome editing or by depleting their transcripts using Cas13d.*

We fully agree that further investigation of the role of NMD factors, such as UPF1 and UPF3a/b, is an important direction for further exploration. Addressing this question thoroughly will require the use of multiple knockdown and knockout approaches, probably even degron approaches, to target various components of the NMD pathway and untangle potential compensatory effects of their disruption. Thus, addressing this question comprehensively will require extensive mechanistic studies that lie beyond the scope of this manuscript.

10. *Expanded view Figure 1: Authors mentioned that they don't see any difference in the levels of control and ACTG-1 proteins in figure1EV A, while to rule out any doubt, authors could crop the gels to highlight only bands of ACTG1 and respective Y-tubulin bands as shown in figure (fig. 2B).*

We have now added letters to the EV Figure 2 to clarify this point.

Reviewer #3:

This is an interesting submission that further characterizes the mechanism triggering "transcriptional adaptation" (TA). There are two theories put forth: one by the authors' lab (Stainier's), and the other by the Peng lab. This submission further substantiates Stainier's theory that, rather than UPF3a functioning in TA as proposed by Peng, RNA degradation (presumably via RNA decay products) formed either in the cytoplasm or, as shown here, in the nucleus, upregulate not only a paralogous gene(s) but the gene from which the RNA decay products derive. Other new findings include assays for open chromatin: open chromatin for the gene upregulated by TA is evident for TA due to a PTC-containing mRNA

that is degraded by NMD but not an mRNA degraded by CRISPR-Cas9 (the significance of which remains unclear - an explanation/comment from the authors would be helpful. This submission moves the field forward in a creative way. It also raises concerns about interpreting CRISPR-Cas9 gene KO data, which is useful for the scientific community to know.

Comments to be addressed by the authors include the following.

We thank the reviewer for their recognition of the importance of this work.

1. Page 5, first paragraph. The meaning of "lacking a normal stop codon" requires clarification. Do the authors mean "containing a PTC"?

In page 5, "lacking a normal stop codon" refers to a non-stop decay (NSD) allele, where an endogenous STOP codon was deleted. We have now clarified this point (lines 89–90).

2. Page 5, first paragraph. No where is "FLD" defined. It must not mean "full-length deletion" since in *Actg1*FLD cells, there were three different lesions, presumably because there are three different alleles for this gene (?), and one lesion leaves behind the last 441 bases of the 3'UTR while another leaves behind 44 bases of the 5'UTR.

We thank the reviewer for pointing out this oversight. 'FLD' stands for 'full locus deletion'. However, since part of the gene is still present in these cells, we now refer to them as *Actg1* 'large deletion (LD)' cells.

3. Page 5, first paragraph. Is RNA produced from the "last 441 bases of the 3'UTR"? This seems to be an important issue to resolve, as it is for the allele leaving behind the first 44 bases of the 5'UTR, which one would think would produce RNA. The authors want to claim that RNA production is required for TA. Thus, they should better characterize, or at the very least make a comment about, the possibility of *Actg1* RNA production in the *Actg1*FLD cells. This reviewer realizes that no mRNA was detected in the steady-state using their RT-qPCR assay for mRNA. However, this may not mean that RNA wasn't produced and subsequently degraded in a way that would not be detected by their RT-qPCR assay.

In a follow-up study, we conducted RNA-seq experiments with poly(A) enrichment for the various alleles and did not detect any transcripts mapped to *Actg1* in the *Actg1*^{LD} mutant cells (data not included in this manuscript), indicating that *Actg1*^{LD} cells yield very few if any *Actg1* transcripts.

4. Page 6, first paragraph. Is TA to the same extent when there is cytoplasmic mRNA degradation without a genomic lesion and when there is a PTC? It would be nice to have a quantitative comparison of Figures 1, 2 and 3. This is done in Figure 3, but without normalization to a cellular RNA (see next comment).

For each gene of interest, its expression levels are first normalized to those of *Hprt*, a housekeeping gene, and then compared among different conditions, i.e., wild-type and mutant cells. Therefore, the fold change between wild-type and mutant cells can be

compared across different experiments, i.e., CRISPR/Cas9, CRISPR/Cas13d, and T3H48-HHR knock-in (see for example the consistent fold change of *Actg2* mRNA and *Actg2* pre-mRNA in Figure 1B and Figure 3B across different experiments).

5. *Figures 1,2 and 3. For the RT-qPCR to be quantitative, the authors need to normalize mRNA and pre-mRNA levels to a cellular RNA. Apparently, this hasn't been done, but it should be.*

All RT-qPCRs were normalized to an internal (housekeeping) control, the *Hprt* gene, and *Hprt* qPCR primer sequences and Ct values are listed in the EV tables.

6. *Page 7, first paragraph. Please change "TA or a TA-like response" to "TA vs. a TA-like response".*

We did not intend to contrast the TA and TA-like responses in this sentence (which has now been modified). It is possible that these processes are one and the same, but additional studies are needed to investigate whether TA and TA-like responses share fundamentally different or similar mechanisms.

7. *Page 8, last paragraph. Interestingly, the extent to which Actg1 pre-mRNA was cleaved by the HHR ribozyme correlated with "Actg2 upregulation". Please specify in the text how cleavage efficiency was assayed.*

The efficiency of *Actg1* pre-mRNA cleavage was inferred from the *Actg1* mRNA levels detected in *Actg1*^{ΔHHR1} and *Actg1*^{ΔHHR2} cells, and is now explained in more detail in the text (page 8, lines 218–221).

References

1. El-Brolosy MA, Kontarakis Z, Rossi A, Kuenne C, Günther S, Fukuda N, Kikhi K, Boezio GLM, Takacs CM, Lai SL, Fukuda R, Gerri C, Giraldez AJ, Stainier DYR. Genetic compensation triggered by mutant mRNA degradation. *Nature*. 2019; 568(7751): 193–197.
2. Ma Z, Zhu P, Shi H, Guo L, Zhang Q, Chen Y, Chen S, Zhang Z, Peng J, Chen J. PTC-bearing mRNA elicits a genetic compensation response via Upf3a and COMPASS components. *Nature*. 2019; 568(7751): 259–263.
3. TODARO GJ, GREEN H. Quantitative studies of the growth of mouse embryo cells in culture and their development into established lines. *J Cell Biol*. 1963; 17(2): 299–313.
4. Beg AA, Baltimore D. An essential role for NF-kappaB in preventing TNF-alpha-induced cell death. *Science*. 1996; 274(5288): 782–784.
5. Lindeboom RG, Supek F, Lehner B. The rules and impact of nonsense-mediated mRNA decay in human cancers. *Nat Genet*. 2016; 48(10): 1112–1118.
6. Kushawah G, Hernandez-Huertas L, Abugattas-Nuñez Del Prado J, Martinez-Morales JR, DeVore ML, Hassan H, Moreno-Sanchez I, Tomas-Gallardo L, Diaz-Moscoso A, Monges DE, Guelfo JR, Theune WC, Brannan EO, Wang W, Corbin TJ, Moran AM,

Sánchez Alvarado A, Málaga-Trillo E, Takacs CM, Bazzini AA, Moreno-Mateos MA. CRISPR-Cas13d Induces Efficient mRNA Knockdown in Animal Embryos. *Dev Cell*. 2020; 54(6): 805–817.

7. Konermann S, Lotfy P, Brindeau NJ, Oki J, Shokhirev MN, Hsu PD. Transcriptome Engineering with RNA-Targeting Type VI-D CRISPR Effectors. *Cell*. 2018; 173(3): 665–676.

Dear Prof. Stainier,

Thank you for the submission of your revised manuscript. We have now received the enclosed reports from the referees that were asked to assess it, and I am happy to say that both support its publication now.

Only a few editorial requests will need to be addressed before we can proceed with the official acceptance of your manuscript:

- Please reduce the number of keywords to 5.
- Please correct the conflict of interest subheading to "Disclosure and Competing Interests Statement"
- The references need to be placed to before the figure legends.
- A callout for Fig 3C is missing, please add.
- The EV table 1 name needs to be corrected to Table EV1, it needs a title or legend in the excel file and it needs to be cited in the ms text as Table EV1.
- The Reagent and Tools table needs to be uploaded as a separate file. The info in Table EV1 can also go directly into the Reagents table.
- The Expanded View Figure 1 needs to be correct to Figure EV1, also in the file itself and ms text and figure legend.
- Materials and Methods should be just Methods

Figure Legends - Comments

- Please note that the exact p values are not provided in the legends of figures 1B, 2A, C, D; 3B, EV1 B. Please provide exact values, as reasonable.
- Please note that the measure of center for the error bars needs to be defined in the legends of figures 1B, 2A, C, D; 3B, EV1 B, EV3.

I would like to suggest a few minor changes to the abstract that needs to be written in present tense:

Transcriptional adaptation (TA) is a cellular process whereby mRNA-destabilizing mutations are associated with the transcriptional upregulation of so-called adapting genes. The nature of the TA-triggering factor(s) remains unclear, namely whether an mRNA-borne premature termination codon or the subsequent mRNA decay process, and/or its products, elicits TA. Here, working with mouse Actg1, we first establish two types of perturbations that lead to mRNA destabilization: Cas9-induced mutations predicted to lead to mutant mRNA decay, and Cas13d-mediated mRNA cleavage. We find that both types of perturbations are effective in degrading Actg1 mRNA, and that they both upregulate Actg2. Notably, increased chromatin accessibility at the Actg2 locus is observed only in the Cas9-induced mutant cells but not in the Cas13d-targeted cells, suggesting that chromatin remodeling is not required for Actg2 upregulation. We further show that ribozyme-mediated Actg1 pre-mRNA cleavage also leads to a robust upregulation of Actg2, and that this upregulation is again independent of chromatin remodeling. Together, these data highlight the critical role of RNA destabilization events as a trigger for TA, or at least a TA-like response.

EMBO press papers are accompanied online by A) a short (1-2 sentences) summary of the findings and their significance, B) 2-3 bullet points highlighting key results and C) a synopsis image that is exactly 550 pixels wide and 200-600 pixels high (the height is variable). The synopsis image should provide a sketch of the major findings, like a graphical abstract. Please note that text needs to be readable at the final size. Please send us this information along with the final manuscript.

Referee #1:

I have looked over the revised MS, along with the Rev criticisms and author responses and am satisfied that this MS is suitable for publication in EMBO Reports.

There are some additional experiments suggested by the Revs, such as determining whether NMD is involved in the TA response, that I think would significantly add to this MS, but I don't regard this as obligatory.

Referee #2:

Thank you to the authors for addressing my questions thoroughly. I now have a better understanding of the limitations and scope of the manuscript. I am satisfied with the authors' responses and would be pleased to see this manuscript published in EMBO Reports.

All editorial and formatting issues were resolved by the authors.

Prof. Didier Stainier
Max Planck Institute for Heart and Lung Research
Department of Developmental Genetics
Ludwigstraße 43
Bad Nauheim 61231
Germany

Dear Prof. Stainier,

I am very pleased to accept your manuscript for publication in the next available issue of EMBO reports. Thank you for your contribution to our journal.
